# The Cholesterol-Modulating Effect of the New Herbal Medicinal Recipe from Yellow Vine (*Coscinium fenestratum* (Goetgh.)), Ginger (*Zingiber officinale* Roscoe.), and Safflower (*Carthamus tinctorius* L.) on Suppressing PCSK9 Expression to Upregulate LDLR Expression in HepG2 Cells

**DOI:** 10.3390/plants11141835

**Published:** 2022-07-13

**Authors:** Tassanee Ongtanasup, Nuntika Prommee, Onkamon Jampa, Thanchanok Limcharoen, Smith Wanmasae, Veeranoot Nissapatorn, Alok K. Paul, Maria de Lourdes Pereira, Polrat Wilairatana, Norased Nasongkla, Komgrit Eawsakul

**Affiliations:** 1School of Medicine, Walailak University, Nakhon Si Thammarat 80160, Thailand; tassanee.on@wu.ac.th (T.O.); thanchanok.li@wu.ac.th (T.L.); 2Research Excellence Center for Innovation and Health Products (RECIHP), Walailak University, Nakhon Si Thammarat 80160, Thailand; nissapat@gmail.com or; 3Division of Applied Thai Traditional Medicine, Faculty of Public Health, Naresuan University, Phitsanulok 65000, Thailand; nuntikap@nu.ac.th; 4Tak Community College, Nong Bua Tai 63000, Thailand; onkamon@takcc.ac.th; 5School of Allied Health Sciences, Walailak University, Nakhon Si Thammarat 80160, Thailand; smith.wa@wu.ac.th; 6School of Allied Health Sciences, World Union for Herbal Drug Discovery (WUHeDD), Nakhon Si Thammarat 80160, Thailand; 7School of Pharmacy and Pharmacology, University of Tasmania, Hobart, TAS 7001, Australia; alok.paul@utas.edu.au; 8CICECO—Aveiro Institute of Materials, Department of Medical Sciences, University of Aveiro, 3810-193 Aveiro, Portugal; mlourdespereira@ua.pt; 9Department of Clinical Tropical Medicine, Faculty of Tropical Medicine, Mahidol University, Bangkok 10400, Thailand; polrat.wil@mahidol.ac.th; 10Department of Biomedical Engineering, Faculty of Engineering, Mahidol University, Nakhon Pathom 73170, Thailand; norased.nas@mahidol.ac.th

**Keywords:** cholesterol-lowering, PCSK9, *C. tinctorius*, *C. fenestratum*, *Z. officinale*, molecular docking, chemical constituents

## Abstract

PCSK9 is a promising target for developing novel cholesterol-lowering drugs. We developed a recipe that combined molecular docking, GC-MS/MS, and real-time PCR to identify potential PCSK9 inhibitors for herb ratio determination. Three herbs, *Carthamus tinctorius*, *Coscinium fenestratum*, and *Zingiber officinale,* were used in this study. This work aimed to evaluate cholesterol-lowering through a PCSK9 inhibitory mechanism of these three herbs for defining a suitable ratio. Chemical constituents were identified using GC-MS/MS. The PCSK9 inhibitory potential of the compounds was determined using molecular docking, real-time PCR, and Oil red O staining. It has been shown that most of the active compounds of *C. fenestratum* and *Z. officinale* inhibit PCSK9 when extracted with water, and *C. fenestratum* has been shown to yield tetraacetyl-d-xylonic nitrile (27.92%) and inositol, 1-deoxy-(24.89%). These compounds could inhibit PCSK9 through the binding of 6 and 5 hydrogen bonds, respectively, while the active compound in *Z. officinale* is 2-Formyl-9-[.beta.-d-ribofuranosyl] hypoxanthine (4.37%) inhibits PCSK9 by forming 8 hydrogen bonds. These results suggest that a recipe comprising three parts *C. fenestratum*, two parts *Z. officinale*, and one part *C. tinctorius* is a suitable herbal ratio for reducing lipid levels in the bloodstream through a PCSK9 inhibitory mechanism.

## 1. Introduction

Blood cholesterol levels of total cholesterol and low-density lipoprotein (LDL) cholesterol are both major risk factors for coronary heart disease (CHD). Reduced total and LDL cholesterol levels have been shown to decrease the risk of coronary heart disease.

The most given lipid-lowering drug is statins, which potently inhibit 3-hydroxy-3-methylglutaryl-coenzyme A (HMG-CoA) reductase, the enzyme that decreases the biosynthesis of cholesterol [1,2,3]. This results in intracellular cholesterol depletion and subsequent upregulation of low-density lipoprotein receptors (LDLRs) expression on hepatocytes and enhanced clearance of LDL from blood circulation via the sterol regulatory element-binding protein (SREBP) pathway. Additionally, proprotein convertase subtilisin kexin type 9 (PCSK9), a member of the subtilisin-related serine protease family, has been identified as a critical regulator of low-density lipoprotein (LDL) metabolism, and inhibitors of PCSK9 are currently being investigated for their ability to lower circulating LDL via binding to its epidermal growth factor-like repeat (EGF-A) of LDLR [4,5,6]. Secreted PCSK9, a domain found in hepatocytes, binds to LDLR and promotes its lysosomal degradation in cells [7,8].

PCSK9 deficiency leads to a more significant number of cell surface LDLRs, and enhanced hepatic LDLR expression leads to improved plasma LDL clearance, protecting against cardiovascular disease (CVD). As a result, finding a new antihyperlipidemic drug that targets PCSK9 expression is a top priority in antihyperlipidemic research. Reducing PCSK9 transcription is a potential technique for lowering LDL. Thus, we set out to find a new recipe that inhibits PCSK9 transcription to promote plasma cholesterol-reduction effects via their effect on LDLR transcription. The new herbal recipe that induced LDLR expression may be a useful technique for treating hyperlipidemia. Complementary and alternative medicine has been utilized to control cholesterol levels and improve heart health; therefore, increasing LDLR expression from herbal drugs might be a useful antihyperlipidemic method. In addition, the use of various herbs as medicinal compounds will help to improve the effectiveness of the treatment.

Yellow vine (*Coscinium fenestratum* (Goetgh.)), commonly called ‘tree turmeric’, belongs to the Menispermaceae family and is a medicinally significant dioecious endangered liana [9] found in Vietnam, Singapore, Sri Lanka, and Thailand [10]. The stem and root of *C. fenestratum* are used in traditional Chinese medicine [9]. Berberine (isoquinoline alkaloids), dropalmatine, crebanine, jatrorrhizine, palmitic acid, oleic acid, and saponin have all been isolated from *C. fenestratum* [11]. These molecules possess various pharmacological effects, including anti-diabetic, anti-inflammatory, thermogenic, and antimicrobial activities [12]. Additionally, multiple studies [13,14,15] suggest berberine’s usefulness in decreasing blood lipids. However, the usage of *C. fenestratum* for cholesterol reduction has not been explored.

*Ginger* (*Zingiber officinale* Roscoe), most commonly known as ginger, is a spice and flavoring ingredient used in cuisines worldwide [16]. For thousands of years, it has been used as a spice and for medicinal purposes. Its usage is attested in ancient Sanskrit and Chinese manuscripts, as well as in Arabic, Roman, and Greek medical literature [17]. *Z. officinale* is regarded as a promising medication in Ayurveda due to its efficacy as a digestive stimulant, antiasthmatic, and rubefacient [18]. It is cultivated commercially in India, China, Thailand, Australia, South Africa, and Mexico. Antioxidant activity [19,20,21] has been reported in vitro for *Z. officinale* aqueous and organic solvent extracts. A combination of *Z. officinale* and garlic [17] was proven to have hypoglycaemic and hypolipidemic effects in albino rats. The previous research [22] has demonstrated that ethanolic *Z. officinale* extract has considerable antihypercholesterolemic action in cholesterol-fed rabbits. It should be emphasized that *Z. officinale*’s efficacy in lowering cholesterol levels is favorable and that its usefulness should be investigated when paired with other herbs.

Safflower (*Carthamus tinctorius* L.) is an oil-producing crop that belongs to the Compositae or Asteraceae family. In Thailand, it is called Kamfoi, whereas, in China, it is called zang hong hua. *C. tinctorius* is a multifunctional crop that has been produced in Thailand and other areas of the world for generations for a variety of purposes. It is a critical plant since it provides an alternate supply of oil. *C. tinctorius* research and development continue to receive little attention [23]. However, it can grow in a wide variety of environmental conditions with very high yield potential and has a variety of uses for the various plant components. However, some researchers [24,25] have reported that *C. tinctorius* contains linoleic acid, an unsaturated fatty acid, which is widely known and helps decrease blood cholesterol levels.

In addition, all three types of herbs—*C. fenestratum,*
*Z. officinale,* and *C. tinctorius*—contain primary metabolite and secondary metabolite. In addition, each country has a wide range of uses as shown in Table 1.

Although all three herbs have been examined for their lipid-lowering properties, none have been combined to create a lipid-lowering recipe. Therefore, in this study, new formulations from these herbs were investigated for lipid-inhibiting activity through mechanisms such as HMG-CoA, SREBP, PCSK9, and LDLR mRNA levels using molecular docking and in vitro studies. Then, the proportion of herbs in the recipe will be determined to be suitable for reducing lipid in the bloodstream.

## 2. Materials and Methods

### 2.1. Materials

#### 2.1.1. Cell Line, Chemicals, and Computer Software

Human hepatocellular carcinoma (HepG2) was purchased from ATCC (Manassas, VA, USA). It was cultivated in Dulbecco’s modified Eagle’s medium (CAS No. 11965118) with 10% fetal bovine serum (CAS No. 10270), 1% PenStrep (CAS No. 15140122), and 3.7 g/L sodium bicarbonate (CAS No. 144-55-8). Filtration of the culture media was performed using a 0.22 m cellulose acetate membrane (CAS No. 11107-25-N). Cells were detached for quantification using 0.25% trypsin-EDTA (CAS No. 25200072; Gibco, Waltham, MA, USA), followed by 0.4% trypan blue staining for cultivated cell viability (CAS No. 15250061). Thiazolyl blue tetrazolium bromide (MTT, CAS No. 298-93-1) and dimethylsulfoxide (DMSO) were used to determine the viability of cells (CAS No. 67-68-5).

Oil red O was purchased from Sigma in the United States of America (CAS No. 1320-06-5) and dissolved in a stock solution by adding 100 mg oil red O to 20 mL100% isopropanol (CAS No. 67-63-0). Prior to staining, a working solution of Oil red O was made by diluting three parts stock solution with two parts DI water. This working solution was filtered using Whatman paper 42. (CAS Number 1442-110).

AutoDock 1.5.6, Python 3.8.2, MGLTools 1.5.4, Discovery Studio-2017, ArgusLab 4.0.1, ChemSketch, Avogadro, and OpenBabel were used to perform molecular docking. The research was conducted by examining the system parameters specified in the software specifications. Processor: Intel Xeon-E5-2678v3 12C/24T CPU @ 2.50 GHz–3.10 GHz, system memory: 32 GB DDR4-2133 RECC, graphics processing unit: VGA GTX 1070 TI 8G, operating system type: 64-bit, with Windows 10 as the operating system.

#### 2.1.2. Herb Material

In August 2021, these three plants were obtained from Thailand’s Vejponggosot pharmaceutical company: *C. tinctorius*, *C. fenestratum*, and *Z. officinale*. The Thai Traditional Medicine Herbarium, Department of Thai Traditional and Alternative Medicine, Bangkok, Thailand, has deposited these herbs. The voucher specimen numbers for *C. tinctorius*, *C. fenestratum*, and *Z. officinale* are TTM-c No. 1000705, TTM-c No. 1000703, and TTM-c No. 1000704, respectively.

### 2.2. Extraction and Isolation

Plant materials were washed and dried at 50 °C until reaching a stable weight, then ground into a powder material and prepared for extraction method.

#### 2.2.1. Water Extraction

The powdered herb (400 g) was mixed with 1 L of warm deionized water. On a hot plate, the herb solution was heated to 100 °C for 15 min. Another 1000 mL of hot water was added to the solution because the herb absorbed the water. The final solution was boiled until only one-third of the solution remained. Prior to freeze-drying, the solution was filtered using Whatman No. 1 filter paper and stored at −20 °C. Freeze-dryer (Eyela FDU-2100, Bohemia, NY, USA) was used to lyophilize the frozen samples.

#### 2.2.2. Ethanol Extraction

Individually, 400 g of *C. fenestratum* stem, *C. tinctorius* flower, and *Z. officinale* rhizomes were extracted with ethanol for three days using the maceration procedure. The filtrate was collected using Whatman No. 1 filter paper and evaporated using a rotary evaporator to obtain a viscous ethanolic extract (Heidolph Basic Hei-VAP ML, Schwabach, Germany). The maceration procedure was then performed twice more. Each herb’s remaining ethanol was evaporated further in a vacuum drying chamber (Binder VD 23, Tuttlingen, Germany) until a stable weight was obtained.

### 2.3. GC-MS/MS Analysis

Scion 436 GC Bruker model performed GC-MS/MS analyses to analyze the material at a 3 mg/mL concentration. The GC-MS/MS separation of the compounds was performed with a 30-m fused silica capillary column (0.25 mm internal diameter, 0.25 µm thickness). The carrier gas was helium gas (99.999 percent) with a constant flow rate of 1 mL/min and an injection volume of 10 µL. (split ratio of 10:1). The injector was heated to 250 °C, while the ion source was heated to 280 °C. The oven temperature was kept at 110 °C for 2 min, increased to 280 °C at 5 °C/min, and then kept isothermal at 280 °C for 9-min, for a total GC run duration of 60 min. The mass analysts by ionization energy of 70 eV with 0.5 s interval scan were designed, with fragments ranging from *m*/*z* 50 to 500 Da. The intake temperature was set to 280 °C, while the source temperature was set at 250 °C. By comparing the average peak area of each component to the total areas, the relative fraction of each component was computed. MS Workstation 8 was used for handling mass spectra and chromatograms. The chemical components were identified using the NIST Version 2.0 library database of the National Institute of Standards and Technology (NIST).

### 2.4. Treatment of HepG2 Cells

The ATCC (Manassas, VA, USA) provided the human hepatocellular carcinoma HepG2 cell line cultured in DMEM supplemented with 10% fetal bovine serum (FBS). The cells were seeded in 96-well plates with 5 × 10^4^ cells/mL in a normal serum medium for 24 h before being changed to DMEM without FBS overnight. For an additional 24 h, cells were treated with extracts of the *C. fenestratum*, *Z. officinale*, and *C. tinctorius*, as well as a recipe of *C. fenestratum* (3 parts), *Z. officinale* (2 parts), and *C. tinctorius* (1 part) extracted with water and ethanol at concentrations ranging from 10 to 400 µg/mL prior to cell viability testing, real-time PCR, and oil red O staining.

#### 2.4.1. Cell Viability Analysis

An MTT assay was used to measure cell viability. Briefly, cells were treated as described above, then incubated for 4 h at 37 °C with a 1 mg/mL MTT solution [43,44]. The purple formazan crystals were dissolved in DMSO when the medium was removed. Cell viability was measured by absorbance at 550 nm of the microplate reader (Metertech M965, Taipei, Taiwan).

#### 2.4.2. Quantitative Reverse Transcription PCR (RT-qPCR) Analysis

The total RNA mini kit (Geneaid, Taipei, Taiwan) was used to isolate total RNA from HepG2 cells. Using an iScript Mastermix (Bio-Rad, Hercules, CA, USA), a quantified 1 µg sample of total RNA was converted to cDNA. The primers for specific genes are listed in Table 2 using the Luna Master Mix. The level of mRNA expression was evaluated using a Quanti-Studio 3 (ThermoFisher, Waltham, MA, USA) according to the manufacturer’s guidelines. To compare the groups, 2^−^^ΔΔ^^CT^ values were used, with GAPDH (glyceraldehyde 3-phosphate dehydrogenase) acting an endogenous control [45].

#### 2.4.3. Oil Red O Staining

Ice-cold PBS rinsed the fasting-induced steatosis in HepG2 cells before being fixed by ice-cold 10% formalin for 30 min. The cells were then rinsed with distilled water and stained for 30 min at room temperature with an Oil Red O working solution to generate stain lipid droplets [46]. An optical microscope was used to study and photograph the cells (Ziess AX10, Carl Zeiss, Jena, Germany). Lipid content was also determined by dissolving Oil red O in isopropanol and measuring using a microplate reader at a wavelength of 500 nm [15].

### 2.5. Molecular Docking

The crystal structures of PCSK9 and HMGCR with the PDB codes 6u26 [47] and 2r4f [48] were utilized. Autodock [49] was used to optimize the protein. The missing hydrogens were inserted throughout the optimization step. The final proteins were given Kollman unified atom charges and solvation parameters. Table 3 shows the grid position and size reflecting the whole protein during the docking process. Following GC-MS/MS analysis, the 3D structures of the top 5 high yielding compounds in *C. tinctorius*, *C. fenestratum*, and *Z. officinale* were chosen for docking, while positive docking controls were Alirocumab [50] and Lovastatin [51] for PCSK9 and HMGCR, respectively. All 3D structures were obtained from PubChem (https://pubchem.ncbi.nlm.nih.gov, accessed on 2 October 2021). All structures were optimized before molecular docking. Open Babel was used to add hydrogen atoms to every structure and all structures were optimized by Arguslab through semi-empirical Parametric Method 3 (PM3). Molecular docking was utilized to explore protein–ligand binding. Arguslab and Autodock were used for this docking study. In the beginning, the Arguslab engine was used for docking. The scoring function was set in default parameters. The accuracy of docking was set to regular. All docking was confirmed with Autodock3 through the Lamarckian genetic technique to ensure reliable results. The following are the optimal autodocking run parameters: number of GA runs: 50; population size: 200; and all other run parameters: default [44,52].

### 2.6. Binding Site Analysis

The structure of the compounds that resulted in lower binding energy to the targeted proteins than the standard drug was taken to visualize the binding characteristics by Discovery Studio. The ligand–protein bindings were presented as 2D and 3D. To identify the structure binding protein, the binding position was compared through CavityPlus (http://www.pkumdl.cn/cavityplus, accessed on 2 November 2021).

### 2.7. Statistical Analysis

The tests were carried out at least three times except molecular docking, and the results are shown as the mean ± standard deviation. SPSS 12.0 (SPSS Inc., Chicago, IL, USA) was used to perform the statistical calculations. The data were evaluated using a one-way ANOVA with Dunnet’s post hoc test, with a *p*-value < 0.05 considered statistically significant.

## 3. Results

### 3.1. GC-MS/MS Analysis

The active compounds of the herbs extracted with water and ethanol were analyzed with GC-MS/MS. In this study, the five most active compounds were selected and classified into three groups: (1) the most common, which were equal to or greater than 10%; (2) the moderately common were those that were greater than 1% but less than 10%; and (3) rare compounds are substances found less than 1% of the time, which are then chosen to study binding by molecular docking. The active compounds in each herb areshown in Table 4, Table 5 and Table 6.

The water extracted from *C. tinctorius* contained about 17 different compounds. Benzofuran, 2,3-dihydro-, with a molecular weight of 120 and a chemical formula of C_8_H_8_O, had the most remarkable peak area percent of 23.24 among the seventeen compounds detected. The second most significant peak was found with 3-Isopropoxy-1,1,1,7,7,7-hexamethyl-3,5,5-tris(trimethylsiloxy)tetrasiloxane, with a molecular weight of 576 and a chemical formula of C_18_H_52_O_7_Si_7,_ with a summative peak area percent of 21.23. The following compounds of 3,4-Dihydroxyphenylglycol, 4TMS derivative; 4H-Pyran-4-one, 2,3-dihydro-3,5-dihydroxy-6-methyl-; and Cyclohexasiloxane, dodecamethyl- had moderate peak area percent. Their respective values of peak area were 8.94, 8.56, and 6.96. C_20_H_42_O_4_Si_4_/458, C_6_H_8_O_4_/144, and C_12_H_36_O_6_Si_6_/444 are their chemical formulas and molecular weights. The compounds with the lowest peak area percent are presented in Table 4 and Appendix A.

The water-extracted *C. fenestratum* contained about 43 different compounds. Tetraacetyl-d-xylonic nitrile with a molecular weight of 343 and a chemical formula of C_14_H_17_NO_9_ had the most significant peak area percent of 27.92 among the forty-three compounds detected. Inositol, 1-deoxy- with a molecular weight of 164 and a chemical formula of C_6_H_12_O_5_, had the second greatest peak, with a summative peak area of 24.89. The following compounds of d-Gala-l-ido-octonic amide, Thieno[2,3-b]pyridine,3-amino-2-(3,3-dimethyl-3,4-dihydroisoquinolin-1-yl)-4,6-dimethyl-, and Megastigmatrienone had moderate peak area percent. Their respective values of summative peak area were 9.94, 5.87, and 5.56. C_8_H_17_NO_8_/255, C_20_H_21_N_3_S/335, and C_13_H_18_O/190 are their chemical formulas and molecular weights. The compounds with the lowest peak area percent are presented in Table 5 and Appendix A.

The water-extracted *Z. officinale* contained about 42 different compounds. With a molecular weight of 194 and a chemical formula of C11H14O3, 2-Butanone, 4-(4-hydroxy-3-methoxyphenyl)- had the greatest peak area percent of 38.21 among the forty-two compounds detected. The following compounds of (1S,5S)-2-Methyl-5-((R)-6-methylhept-5-en-2-yl)bicyclo[3.1.0]hex-2-ene, 1-(4-Hydroxy-3-methoxyphenyl)dec-4-en-3-one, 2-Formyl-9-[.beta.-d-ribofuranosyl]hypoxanthine, (1S,5S)-4-Methylene-1-((R)-6-methylhept-5-en-2-yl)bicyclo[3.1.0]hexane had moderate peak area percent. Their respective values of summative peak area were 9.06, 5.89, 4.37, and 3.77. C_15_H_24_/204, C_17_H_24_O_3_/276, C_11_H_12_N_4_O_6_/296, and C_15_H_24_/204 are their chemical formulas and molecular weights. The compounds with the lowest peak area percent are presented in Table 6 and Appendix A.

The ethanolic extracts of the three herbs are listed in Table 7, Table 8 and Table 9. The substances of the *C. fenestratum* contained mainly Inositol Inositol, 1-deoxy- at 21.46% and Megastigmatrienone, about 12.63%. *Z. officinale* contains approximately 33.27% butan-2-one, 4-(3-hydroxy-2-methoxyphenyl)- and 1-(4-Hydroxy-3-methoxyphenyl)dec-4-en. -3-one about 24.37%. Finally, *C. tinctorius* contains the main compound of 4H-Pyran-4-one, 2,3-dihydro-3,5-dihydroxy-6-methyl- approx. 12.60%.

### 3.2. Determination of Maximum Dose for HepG2

The cytotoxicity of these herbs—*C. fenestratum*, *Z. officinale*, and *C. tinctorius*—extracted with water and ethanol from concentrations of 10–400 µg/mL were investigated in HepG2 cells by MTT assays. The findings revealed that all herbs extracted with water or ethanol at concentrations less than 50 µg/mL were harmless to HepG2 cells (cell viability >80%). In Figure 1, water extraction of the *C. fenestratum*, *Z. officinale*, and *C. tinctorius* at 50 µg/mL resulted in HepG2 cell survival rates of 88.16%, 90.19%, and 97.28%, respectively. Furthermore, ethanol extraction of *C. fenestratum*, *Z. officinale*, and *C. tinctorius* at 50 µg/mL resulted in cell survival of 103.63%, 82.75%, and 102.71%, respectively. As a result, the maximum dosage of those herbs was indicated for further research at 50 µg/mL. From the experiement, it was found that *Z. officinale* extracted with ethanol had the highest toxicity. Concentration values calculated using the fitting curve showed that the maximum concentration of *Z. officinale* extracted with ethanol that made HepG2 cells non-toxicity was 54.16 ± 3.90 µg/mL. In addition, The MTT assay was used to assess the safety of this recipe. It was revealed that a 3:2:1 ratio of *C. fenestratum*, *Z. officinale*, and *C. tinctorius* could be safely used at concentrations up to 100 µg/mL in this recipe.

### 3.3. Effect of the C. fenestratum, Z. officinale, and C. tinctorius on Transcriptional Activity of HMGCR, LDLR, PCSK9, and SREBP2

The previous study [53] on the correlation between SREBP2 and PCSK9 has indicated that inhibiting transcriptional activation of the sterol regulatory element binding protein 2 (SREBP2), which regulates PCSK9, increases LDLR expression, as seen in Figure 2. It was discovered that inhibiting SREBP2 expression enhanced LDLR activation. *C. fenestratum* extracted with water and ethanol has lipid-lowering activity through upregulating hepatic LDLR. Among three herbs with two types of extraction, this study found that the most effective way to upregulate LDLR expression by up to 23.12-fold was to treat with water-extracted *C. fenestratum*, followed by water-extracted *Z. officinale,* which increased the expression of LDLR mRNA by up to 9.09-fold.

From LDLR mRNA, the number of LDLR expressions on the surface of hepatocytes is a significant factor [54]. Water-extracted *C. fenestratum* showed the most significant LDLR mRNA expression in HepG2 cells, followed by ethanol-extracted *C. fenestratum*, water-extracted *Z. officinale*, and ethanol-extracted *C. tinctorius*. The reduction of PCSK9 mRNA expression is the primary cause of LDLR mRNA expression, as seen in Figure 2. Although *Z. officinale*’s potency is less effective at inhibiting PCSK9 than the *C. fenestratum*, *Z. officinale* extract was most effective at suppressing HMGR mRNA expression, as shown in Figure 2. Therefore, the presence of *Z. officinale* in the recipe can reduce the production of lipids from the liver, resulting in lowering blood lipids. In Thai traditional medicine, in addition considering the effectiveness of treatment with main and assistance drugs, it is also essential to add an herb that makes it more appetizing by adjusting the color. Therefore, *C. tinctorius*, which gives it its reddish-orange color and is used as a lipid-lowering herb [55], is used to improve its color.

### 3.4. Effect of Lipid Deposition in HepG2

According to the lipid staining with Oil red O examination, the total lipid in HepG2 cells following treatment with water and ethanol extraction of the *C. fenestratum* was 0.95 and 0.77 folds; *C. tinctorius* was 0.80 and 0.86 folds; *Z. officinale* was 0.78 and 0.73 folds, and the recipe was 0.61 and 0.48 folds, respectively. We found that treating HepG2 cells for 24 h with a recipe containing *C. fenestratum*, *Z. officinale*, and *C. tinctorius* had a strong synergistic effect, causing a significant reduction in lipid deposition when compared to individual herbs. Furthermore, these herbs extracted with ethanol were discovered to play an essential role in lowering the quantity of lipid accumulated in the HepG2 cell. The low lipid accumulation in HepG2 cells was due to the suppression of lipid synthesis, which resulted in a reduction in the quantity of lipid stained in the HepG2 cells.

In this experiment, *Z. officinale* exhibited more significant inhibition of HMGCR mRNA than lovastatin (2.5 times) [56] through 0.51- and 1.34-fold increases in HMGCR mRNA expression in ethanol and water extracts, respectively, compared to the control. In addition, when comparing the HMGCR mRNA inhibition of the extracts with statins, it was found that all herbal extracts inhibited HMGCR mRNA better than all statins. The inhibition value of herbal extracts ranged from 0.52–7.69-fold. The results also compared statins such as simvastatin, pravastatin, fluvastatin, atorvastatin, and rosuvastatin, which can induce HMGCR mRNA expression by up to 15-, 12-, 11-, 9-, and 17-fold in order [56]. The HMGCR mRNA expression found that the three herbal extracts had better properties in inhibiting lipid formation than statins.

Statins have good inhibitory properties in the production of lipids from the liver. Therefore, *Z. officinale* with a mechanism of action that inhibits HMGCR mRNA expression is also effective in inhibiting lipid synthesis. As a result, the lipid accumulation in HepG2 cells was lower than in other herbs, as shown in Figure 3. However, the large amount of lipid accumulation in the HepG2 cells of *C. fenestratum* results from most of the compounds suppressing the PCSK9 expression, which results in increased LDLR expression. However, it has little effect on the expression of HMG-CoA reductase (HMGCR). This causes more lipid to be absorbed into HepG2 cells.

According to Thai traditional knowledge, the recipe composition is divided into three parts: the main drug, the assistance drug, and the servant drug. Therefore, the main drug was classified as the *C. fenestratum* in the highest proportion in this study. After all, it was the effect that needed to absorb lipid to the liver from the bloodstream, followed by *Z. officinale* as an assistance drug because it has properties to inhibit the production of lipid from the liver, and *C. tinctorius* as the servant drug, which helps to adjust the color of the recipe to make it more appetizing.

### 3.5. Molecular Docking for the Top 5 Highest Amounts of the Compound from Each Herb

Figure 4 and Table 10 show that PCSK9 has three pocket-binding sites: strong binding sites, medium binding sites, and low binding sites. Figure 4B,D shows three strong binding sites, one medium binding site, and six low binding sites. Water extraction of *C. fenestratum* including Inositol, 1-deoxy-, Tetraacetyl-d-xylonic nitrile, Megastigmatrienone, and Thieno[2,3-b]pyridine, 3-amino-2-(3,). 3-dimethyl-3,4-dihydroisoquinolin-1-yl)-4,6-dimethyl- binds to PCSK9 at a strong binding site. *Z. officinale* extract with water is 2-Formyl-9-[.beta.-d-ribofuranosyl]hypoxanthine, (1S,5S)-2-Methyl-5-((R)-6-methylhept-5-en-2-. yl)bicyclo[3.1.0]hex-2-ene, 2-Butanone, 4-(4-hydroxy-3-methoxyphenyl)-, 1-(4-Hydroxy-3-methoxyphenyl)dec-4-en-3- one, and (1S,5S)-4-Methylene-1-((R)-6-methylhept-5-en-2-yl)bicyclo[3.1.0]hexane. It was found that it was able to bind the PCSK9 region at the strong binding site. Aqueous *C. tinctorius* extract showed that Cyclohexasiloxane, dodecamethyl- binds to PCSK9 at the low binding site and 3,4-Dihydroxyphenylglycol, 4TMS derivative binds to PCSK9 at the strong binding site.

In conclusion, extracts of *C. fenestratum* and *Z. officinale* with water effectively inhibit PCSK9 at the strong binding site, resulting in the most effective inhibition of PCSK9. It was found that the extract could bind to PCSK9 in multiple pocket-binding sites, resulting in combinational inhibition efficiency [57]. After examining the active compounds in each herb via GC-MS/MS, the constituents of the active compounds were identified. The top five compounds were studied through molecular docking to determine that compounds PCSK9 and HMGCR exhibit protein-binding activities. The molecular docking binding studies showed that the effect was consistent with real-time PCR.

In Table 11, the binding between the active ingredients in the herbal aqueous extract and PCSK9 via Arguslab and Autodock showed that approximately 64.24% *C. fenestratum* including Tetraacetyl-d-xylonic nitrile, Inositol, 1-deoxy-, Thieno[2,3-b]pyridine, 3-amino-2-(3,3-dimethyl-3,4-dihydroisoquinolin-1-yl)-4, 6-dimethyl-, Megastigma-trienone binds the most to PCSK9 as it was able to bind to PCSK9 at a lower binding energy than the Alirocumab (standard drug). In Figure 5, Figure 6 and Figure 7, the highest number of compounds found in *C. fenestratum* are 1) Tetraacetyl-d-xylonic nitrile (27.92%). It strongly binds to PCSK9, forming up to six hydrogen bonds with the amino acids HIS643, VAL644, ARG495, and TRP566. 2) Inositol, 1-deoxy- (24.89%) can bind the PCSK9 with different amino acids compared to Tetraacetyl-d-xylonic nitrile-PCSK9 binding. It can form up to five hydrogen bonds with the amino acids TRP461, ALA649, VAL435, and ASN439. Followed by the main active compounds of *Z. officinale*, including 2-Butanone, 4-(4-hydroxy-3-methoxyphenyl)-, (1S,5S)-2-Methyl-5-((R)). -6-methylhept-5-en-2-yl)bicyclo[3.1.0]hex-2-ene, 1-(4-Hydroxy-3-methoxyphenyl)dec-4-en-3-one, 2-Formyl- 9-[.beta.-d-ribofuranosyl]hypoxanthine, (1S,5S)-4-Methylene-1-((R)-6-methylhept-5-en-2-yl)bicyclo[3.1.0]hexane binds to PCSK9 because the number of active compounds that can bind to PCSK9 is 61.3%, and it was found that *Z. officinale* contains only 1 compound, and 2-Formyl-9-[.beta.-d-ribofuranosyl]hypoxanthine contained only 4.37% of *Z. officinale* extract to form a high 8-position hydrogen bond with the amino acids TRP461, LEU436, ASP360, ARG458, ALA649, ASP651, and THR469. The compound number of *C. fenestratum* extracts that can bind to PCSK9 is larger than the compound number of *Z. officinale* extracts, resulting in the water extract of *C. fenestratum* having a better inhibition effect than *Z. officinale*. In comparison, *C. tinctorius* ‘s active compounds have poor binding to PCSK9 because it contains only two compounds: 3,4-Dihydroxyphenylglycol, 4TMS derivative (8.94%), and Cyclohexasiloxane, dodecamethyl-(6.96%), which were found to total just 15.9%, resulting in poor inhibition of PCSK9. These compounds formed very few hydrogen bonds with PCSK9 binding compared to the two herbs mentioned above. Therefore, the preparation of the traditional recipe [58] suggested that the main drug with an excellent inhibitory effect in the highest proportion is *C. fenestratum* (3 parts), the assisting drug (2 parts) is *Z. officinale*, and the flavorful herb is *C. tinctorius* (1 part).

In Table 12, the binding of active compounds in herbs extracted with ethanol and PCSK9 studied via Arguslab and Autodock showed that compounds of *Z. officinale* had a 71.62% inhibitor to PCSK9 as compared to *C. fenestratum* containing a total active inhibitor of 47.04%, thus resulting in better inhibition to PCSK9 of *Z. officinale* than *C. fenestratum* when extracted with ethanol. The results are consistent with the real-time PCR results. It was concluded that the most effective inhibitor of PCSK9 was herbal extracts in water because in water extracts, it was found that the active compounds in *C. fenestratum* and *Z. officinale* extracts are 64.24% and 61.3%, respectively. By comparison, the herb extracts in ethanol provide active *C. fenestratum* and *Z. officinale* compounds at 47.04% and 71.62%, respectively. Therefore, when combining the active compounds for PCSK9 inhibition, *C. fenestratum* and *Z. officinale* suggest the best extraction in the water extract. In addition, studies on the inhibition of HMGCR through Arguslab and Autodock showed that no herbal extract was more effective at inhibiting HMGCR than lovastatin (positive control). The study in Table 13 and Table 14 found that most of the compounds in *Z. officinale* had good efficacy in inhibiting HMGCR compared to extracts of *C. fenestratum* and C. tinctorius. The results are consistent with the effect of real-time PCR. Therefore, the mechanism of HMGCR affecting lipid formation can be best suppressed with *Z. officinale* extract and is classified as an assistance drug in this recipe.

Table 15 and Table 16 show that the ethanol extract of *Z. officinale* had a better binding effect on SREBP2 than the aqueous extract. Four substances of ethanol extraction of *Z. officinale*, consisting of (1) 1,3-Cyclohexadiene, 5-(1,5-dimethyl-4-hexenyl)-2-methyl-, [S-(R*,S*). ]-, (2) 1-(4-Hydroxy-3-methoxyphenyl)dodec-4-en-3-one, (3) (E)-1-(4-Hydroxy-3-methoxyphenyl)dec-3-en-5-one, and (4) 1-(4-Hydroxy-3-methoxyphenyl)dec-4-en-3-one *Z. officinale* with aqueous extract were less binding to SREBP2 because there were only three active substances with energy binding less than −10 kcal/mol: (1) (1S,5S)-2-Methyl-5-((R)-6-methylhept-5-en-2-yl)bicyclo[3.1.0]hex-2-ene, (2) 1-(4- Hydroxy-3-methoxyphenyl)dec-4-en-3-one, and (3) (1S,5S)-4-Methylene-1-((R)-6-methylhept-5-en-2-yl)bicyclo[3.1.0]hexane.

Interestingly, the aqueous extract of *C. fenestratum* contained only one substance, megastigmatrienone. The binding of SREBP2 was lower than −10 kcal/mol, but the inhibition efficiency was higher in the ethanol extraction. This is because there are two active substances that caFn inhibit SREBP2 using energy below −10 kcal/mol: Cyclopropanetetradecanoic acid, 2-octyl-, methyl ester and Megastigmatrienone. The results are also consistent with RT-PCR regarding the expression of SREBP2.

In conclusion, the extracts with the best SREBP2 inhibition were ranked from highest to lowest efficiency. In the following order, *Z. officinale*, *C. fenestratum,* and *C. tinctorius* extracts were extracted, respectively, and it was found that the ethanol extract had a better inhibitory effect than the aqueous extract.

## 4. Discussion

High levels of cholesterol are a significant risk factor for atherosclerosis and cardiovascular disease. Reducing the blood lipid profile may aid in the treatment of high levels of cholesterol-related diseases and disorders, including metabolic syndrome. Statins are medications that can lower cholesterol in a blood vessel and should be taken by most individuals. However, even after taking statins, the lipids in the blood in some individuals remained high [59]. Statins merely enhance the LDLR expression. LDLR destruction stays high if PCSK9 expression is still high [7]. Even though PCSK9 inhibition is beneficial for lipid reduction, the striking benefit achieved with only statin treatments in patients with a wide range of cholesterol levels cannot be attributed to their cholesterol-lowering effect. Therefore, inhibiting PCSK9 expression is crucial for improving lipid reduction.

In this study, the lowering cholesterol activity of three plants, *C. tinctorius*, *C. fenestratum*, and *Z. officinale*, as well as the potential molecular mechanisms involved in their lowering cholesterol activity, were investigated in the human liver cell line HepG2 by using molecular docking and RT-qPCR. Furthermore, we proved that combining these plants by making three parts *C. fenestratum* (primary herb), two parts *Z. officinale* (support herb), and one part *C. tinctorius* (coloring herb) significantly reduced lipid accumulation in hepatocytes by investigating Oil red O staining.

According to these findings, water-extracted *C. fenestratum* was the most effective at downregulating PCSK9 mRNA in HepG2 cells, followed by ethanol-extracted *Z. officinale*, water-extracted ginger, and water-extracted *C. tinctorius*. PCSK9 expression was reduced, which increased LDLR expression. Water-extracted *C. fenestratum* exhibited the most significant induction of LDLR expression, followed by water-extracted *Z. officinale* and water-extracted *C. tinctorius*. Further GC-MS/MS analysis of active compounds for these herbs revealed that excellent inhibition of lipid deposition depended on the efficacy of binding to target proteins and the number of chemical compounds present in the herb. Studies have shown that the highest number of compounds found in the *C. fenestratum* are the following: (1) Tetraacetyl-d-xylonic nitrile (27.92%). It binds strongly to PCSK9, forming up to six hydrogen bonds with the amino acids HIS643, VAL644, ARG495, and TRP566. (2) Inositol, 1-deoxy- (24.89%) can bind the PCSK9 with different amino acids compared to Tetraacetyl-d-xylonic nitrile-PCSK9 binding. It can form up to five hydrogen bonds with the amino acids TRP461, ALA649, VAL435, and ASN439. *Z. officinale* contains only 1 compound, 2-Formyl-9-[.beta.-d-ribofuranosyl]hypoxanthine, which contained only 4.37% of *Z. officinale* extract to form a high 8-position hydrogen bond with the amino acids TRP461, LEU436, ASP360, ARG458, ALA649, ASP651, and THR469. Finally, *C. tinctorius*. *C. tinctorius*’s active compounds have poor binding to PCSK9 because it contains only two compounds: 3,4-Dihydroxyphenylglycol, 4TMS derivative (8.94%), and Cyclohexasiloxane, dodecamethyl-(6.96%), which were found to total just 15.9%, resulting in poor inhibition of PCSK9. These compounds formed very few hydrogen bonds with PCSK9 binding. *C. fenestratum* is the best PCSK9 inhibitor because of its high binding to the target protein and its high active compounds, followed by *Z. officinale*, which has a better PCSK9 inhibitor than the *C. fenestratum*. However, the low content of active compounds resulted in less efficacy of *Z. officinale* in inhibiting PCSK9. Finally, *C. tinctorius* was the least effective in inhibiting PCSK9 because of its fewer active compounds and poorer binding capacity than the herbs, as mentioned earlier. From the study results, an herbal recipe for reducing lipid has been designed by using the knowledge of Thai traditional medicine [58] to set the drug recipe as the main drug, which is the drug that has the highest efficiency in inhibiting lipid with the highest ratio. This recipe is three parts *C. fenestratum*. An assistance drug is a drug that will increase the efficiency of the main drug to reduce lipid with a lesser ratio. This recipe is two parts *Z. officinale*, and a colorant drug is used for adding applicability to the recipe with the lowest ratio. One part of *C. tinctorius* was added to this recipe. This recipe was tested for lipid reduction efficacy using HepG2 cells. It was found that this recipe could reduce lipid accumulation better than using the herb alone. Therefore, this is the world’s first herbal recipe that helps reduce lipid through PCSK9 inhibition.

To clarify the substance structure and biological activity, the study found that the main inhibitors of PCSK9 were tetraacetyl-d-xylonic nitrile (27.92 percent) from *C. fenestratum*, and 2-Formyl-9-[.beta.-d-ribofuranosyl]hypoxanthine (4.37%) from *Z. officinale*. The study of Structure-Activity Relationship (SAR) is available through the website: http://way2drug.com/PassOnline/predict.php. The structure of a substance with a Pa value greater than 0.7 indicates that the substance can be developed as a drug for the treatment of such diseases [60]. The composition analysis of *C. fenestratum* showed that tetraacetyl-d-xylonic nitrile (CC(=O)OCC(C(C(C(=O)C#N)OC(=O)C)OC(=O) C)OC(=O)C) showed very good properties as a lipid metabolism regulator. Pa = 0.822 and *Z. officinale* containing 2-Formyl-9-[.beta.-d-ribofuranosyl] hypoxanthine (C1=NC2=C(N1C3C(C(C(O3)CO)O)O)N=C(NC2=O)C=O) has very good lipotropic properties, with Pa = 0.870. The aforementioned data clearly show that the extracts of *C. fenestratum* and *Z. officinale* have good properties in lowering lipid levels.

Although extractions involve many methods and a variety of solvents, the water and ethanol extraction methods are traditional and easy to implement. The introductions of tetraacetyl-d-xylonic nitrile and 2-Formyl-9-[.beta.-d-ribofuranosyl] hypoxanthine were assessed according to the solubility calculation with SWISSADME, tetraacetyl-d-xylonic nitrile had Log S (ESOL)[61], Log S (Ali) [62], and Log S (SILICOS-IT) [63] as −0.94, −2.22, and −0.74, respectively. The values showed that the compound had high water solubility. Formyl-9-[.beta.-d-ribofuranosyl] hypoxanthine, the values of Log S (ESOL), Log S (Ali), and Log S (SILICOS-IT) were −0.90, −1.24, and 0.20, respectively, refer to high water solubility. From the calculation of solubility, Formyl-9-[.beta.-d-ribofuranosyl] hypoxanthine has slightly better water solubility than tetraacetyl-d-xylonic nitrile. As a result, both compounds with PCSK9 inhibitory activity were better extracted with water than ethanol, consistent with the results of the GC-MS/MS study that found tetraacetyl-d-xylonic nitrile in 27.92% water extraction while extracting only 9.47% with ethanol. Moreover, 2-Formyl-9-[.beta.-d-ribofuranosyl] hypoxanthine was extracted with a 4.37% yield in water, while there are no compounds found in ethanol extraction.

## 5. Conclusions

In conclusion, for screening PCSK9 inhibitors from three plants, *C. tinctorius*, *C. fenestratum*, and *Z. officinale*, an efficient technique incorporating molecular docking, RT-qPCR test, in vitro cytotoxicity, and Oil red O staining assay was devised. Two chemicals had a high yield from *C. fenestratum* based on GC-MS/MS detection: tetraacetyl-d-xylonic nitrile (27.92 percent) and Inositol, 1-deoxy- (24.89 percent). These compounds could inhibit PCSK9 strongly through the binding of 6 and 5 hydrogen bonds, respectively, while the active compound in *Z. officinale* is 2-Formyl-9-[.beta.-d-ribofuranosyl] hypoxanthine (4.37%), which inhibits PCSK9 by forming 8 hydrogen bonds. According to our findings, we may utilize a formula consisting of three parts *C. fenestratum* (primary herb), two parts *Z. officinale* (assistance herb), and one part *C. tinctorius* (servant herb) to define a reasonable herbal ratio for the intervention and prevention of PCSK9-related disorders in the future. Furthermore, because of targeted screening and precise analysis, this technique is expected to be used for a broader range of applications, such as fast screening of active components from herbs, and improving herb ratios in alternative medicine.

## Figures and Tables

**Figure 1 plants-11-01835-f001:**
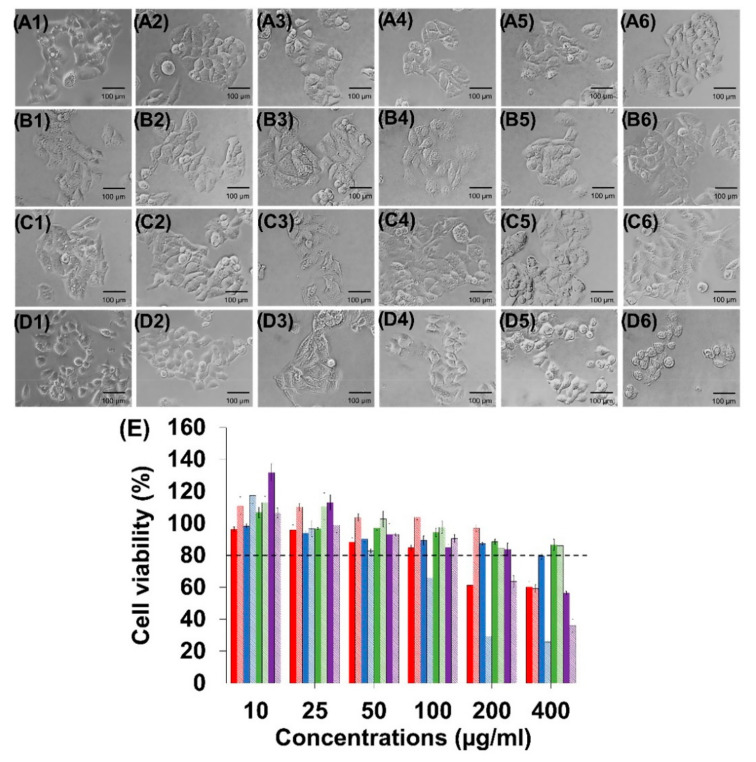
Cell survival and cytotoxicity testing of the HepG2 cells. (**A**) Morphology was exposed to different concentrations ((**A1**): 10 µg/mL; (**A2**): 25 µg/mL; (**A3**): 50 µg/mL; (**A4**): 100 µg/mL; (**A5**): 200 µg/mL; and (**A6**): 400 µg/mL) of *C. fenestratum* from water extraction. (**B**) Morphology was exposed to different concentrations ((**B1**): 10 µg/mL; (**B2**): 25 µg/mL; (**B3**): 50 µg/mL; (**B4**): 100 µg/mL; (**B5**): 200 µg/mL; and (**B6**): 400 µg/mL) of *Z. officinale* from water extraction. (**C**) Morphology was exposed to different concentrations ((**C1**): 10 µg/mL; (**C2**): 25 µg/mL; (**C3**): 50 µg/mL; (**C4**): 100 µg/mL; (**C5**): 200 µg/mL; and (**C6**): 400 µg/mL) of *C. tinctorius* from water extraction. (**D**) Morphology was exposed to different concentrations (**D1**): 10 µg/mL; (**D2**): 25 µg/mL; (**D3**): 50 µg/mL; (**D4**): 100 µg/mL; (**D5**): 200 µg/mL; and (**D6**): 400 µg/mL) of medicinal recipe containing *C. fenestratum*: *Z. officinale*: and *C. tinctorius* extracted with water in a ratio of 3:2:1. (**E**) MTT assay of HepG2 cells treated with different concentrations of the *C. fenestratum* (Water extract: red bar and Ethanolic: red stripes), *Z. officinale* (water extract: blue bar and ethanolic: blue stripes), *C. tinctorius* (water extract: green bar and ethanolic: green stripes), and Recipe (water extract: purple bar and ethanolic: purple stripes.

**Figure 2 plants-11-01835-f002:**
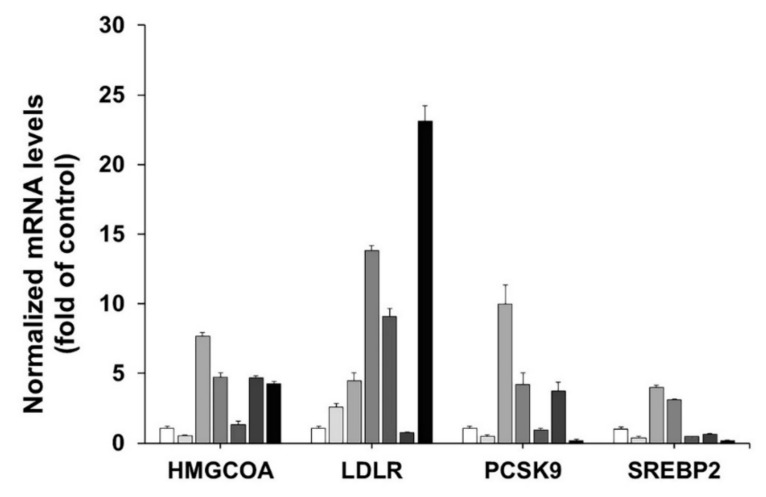
Effects of aqueous and ethanolic extract of *Z. officinale*, *C. tinctorius*, and *C. fenestratum* on mRNA expression levels. The bar graphs go from white (left) to black (right), indicating the control (white), the ethanolic extract of *Z. officinale* (light gray), *C. tinctorius* (medium gray), *C. fenestratum* (dark gray), the water extract of *Z. officinale* (light black), *C. tinctorius* (medium black), and *C. fenestratum* (black) respectively.

**Figure 3 plants-11-01835-f003:**
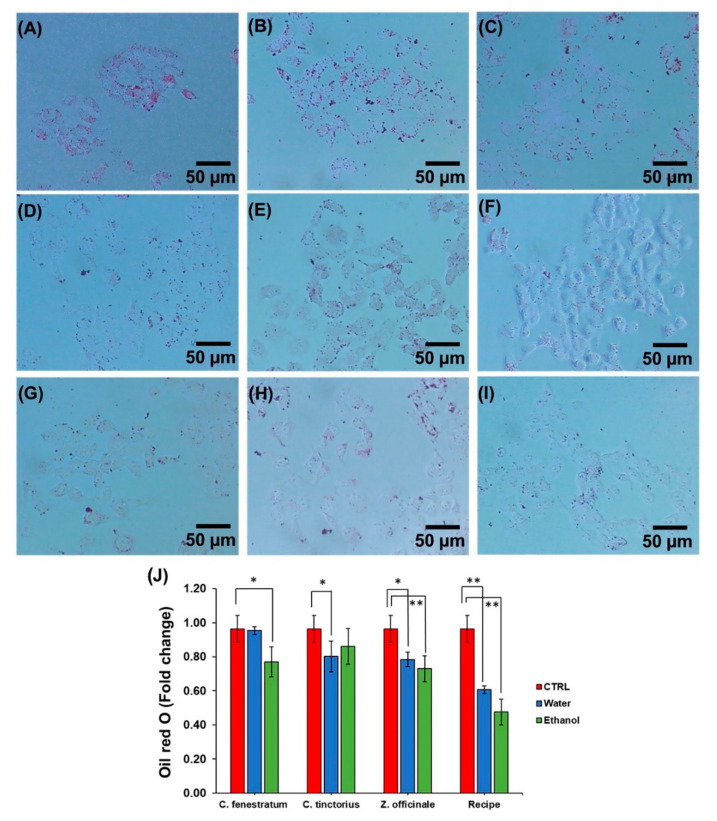
Effects of Oil red-O staining in HepG2 and examined using an inverted microscope. Oil red-O staining of HepG2 was incubated with water extract of (**B**) *C. fenestratum*, (**D**) *C. tinctorius*, (**F**) *Z. officinale*, and (**H**) recipe and Ethanolic extract of (**C**) *C. fenestratum*, (**E**) *C. tinctorius*, (**G**) *Z. officinale*, and (**I**) recipe compared to without treatment as (**A**) control. (**J**) Quantification of lipid accumulation by extracting oil red-O with isopropanol and measuring the OD of extract at 500 nm.

**Figure 4 plants-11-01835-f004:**
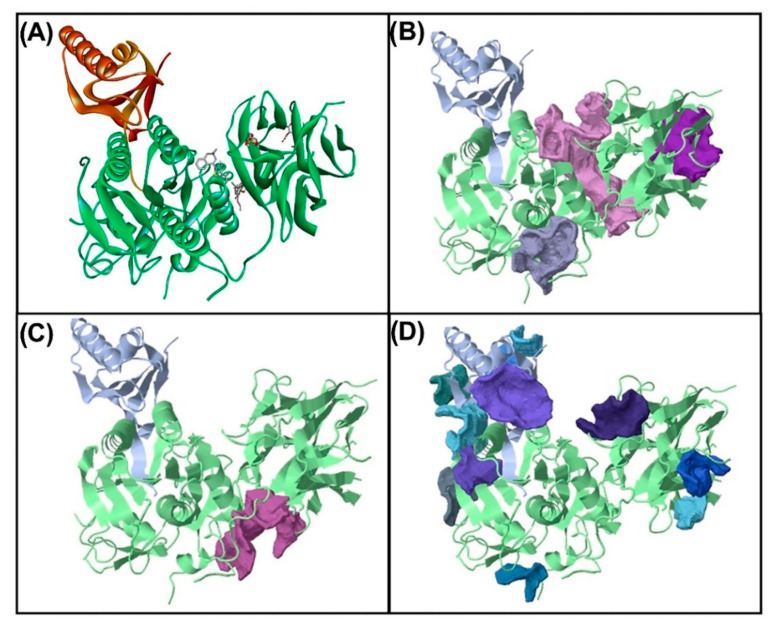
The pocket binding sites of the (**A**) PCSK9 protein at (**B**) high, (**C**) medium, and (**D**) low binding affinity was analyzed with CavityPlus (http://www.pkumdl.cn/cavityplus, accessed on 2 November 2021).

**Figure 5 plants-11-01835-f005:**
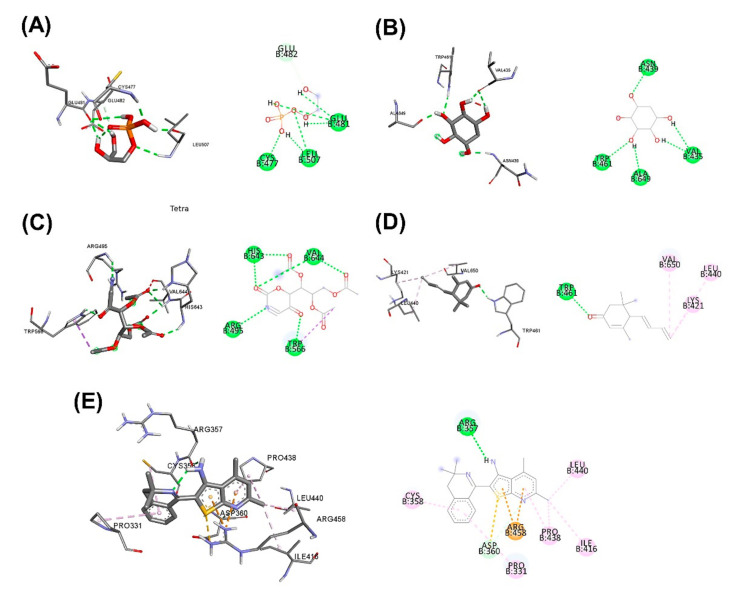
3D (LHS) and 2D (RHS) Molecular docking pose visualization showing water extraction of *C. fenestratum*: (**A**) Alirocumab, (**B**) Inositol, 1-deoxy-, (**C**) Tetraacetyl-d-xylonic nitrile, (**D**) Megastigmatrienone, (**E**) Thieno[2,3-b]pyridine, 3-amino-2-(3,3-dimethyl-3,4-dihydroisoquinolin-1-yl)-4,6-dimethyl- interactions with PCSK9.

**Figure 6 plants-11-01835-f006:**
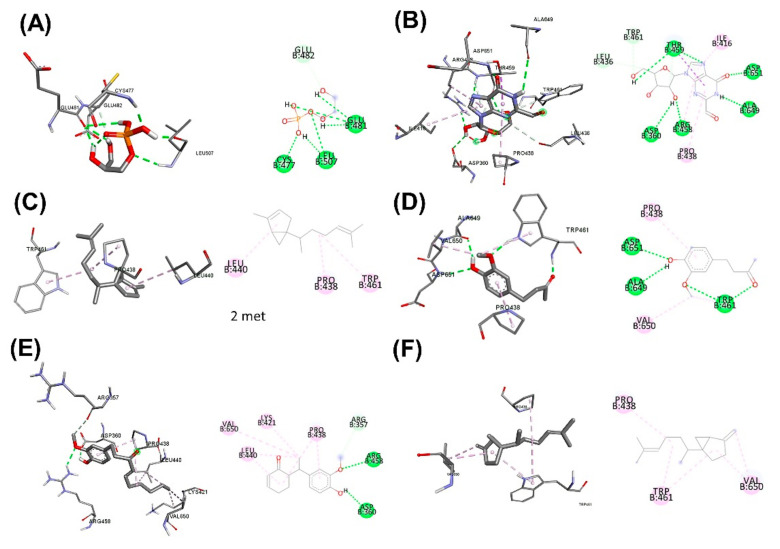
3D (LHS) and 2D (RHS) Molecular docking pose visualization showing water extraction of *Z. officinale*: (**A**) Alirocumab, (**B**) 2-Formyl-9-[.beta.-d-ribofuranosyl]hypoxanthine, (**C**) (1S,5S)-2-Methyl-5-((R)-6-methylhept-5-en-2-yl)bicyclo[3.1.0]hex-2-ene, (**D**) 2-Butanone, 4-(4-hydroxy-3-methoxyphenyl)-, (**E**) 1-(4-Hydroxy-3-methoxyphenyl)dec-4-en-3-one, (**F**) (1S,5S)-4-Methylene-1-((R)-6-methylhept-5-en-2-yl)bicyclo[3.1.0]hexane interactions with PCSK9.

**Figure 7 plants-11-01835-f007:**
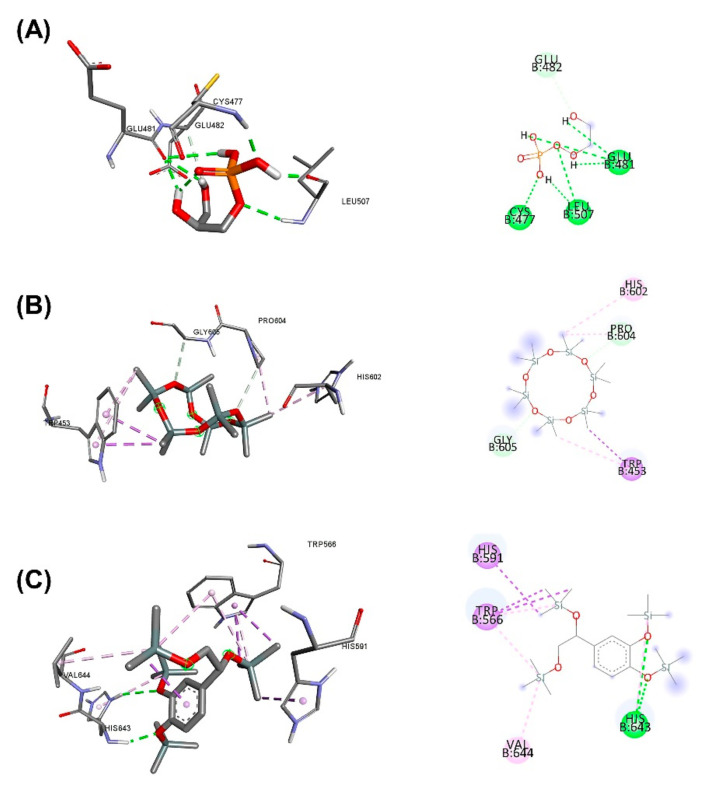
3D (LHS) and 2D (RHS) Molecular docking pose visualization showing water extraction of *C. tinctorius*: (**A**) Alirocumab, (**B**) Cyclohexasiloxane, dodecamethyl-, (**C**) 3,4-Dihydroxyphenylglycol, 4TMS derivative interactions with PCSK9.

**Table 1 plants-11-01835-t001:** Primary and secondary compounds derived from plants and their therapeutic uses in different country.

Scientific Name	Primary Metabolite	Secondary Metabolite	Uses of Plants in Different Countries	Preparations/Therapeutic Uses
*Z. officinale*	carbohydrate, lipids, amino acids, cinnamic acid, and vitamins [26]	oleoresin, phenolics, zingiberene, gingerols, shogaols, aromatic alcohol, and terpenoids [27]	It is distributed all over the world, such as in European countries, America, China, Japan, and India [28] with the following benefits: reducing effect on blood lipids [29] curing heart problems, treating stomach upset, diarrhea, headaches, and cough or nausea [30] treating digestive problems [30] Antibacterial agent [31] Chemopreventive effect [32] vomiting in motion sickness [33]	Use both fresh and dried preparation of rhizome for medicinal use [30] Steam distillation/supercritical CO_2_ extraction for essential oil [31]
*C.fenestratum*	carbohy-drate, lipids, amino acids, and vitamins [34]	alkaloids, tannins, saponins, flavonoids, phenolic compounds [35]	It is distributed all over the world, such as in Sri Lanka, India, and Thailand with the following benefits: antidiabetic, diuretic, cholesterol lowering, anticancer, anti-inflammatory, antifungal, antihelmintic, antioxidant, and antimicrobial effects [36,37]	Use stem and dried preparation with solvent extractions such as Ethanol [38] Methanol [39] Water [40]
*C. tinctorius*	formic acid, acetic acid, succinic acid, glucose, fructose, asparagine, proline, alanine, glutamine, valine, uridine, trigonelline, and choline [41]	saffloquinoside C, saffloquinoside A, anhydrosafflor yellow B, rutin, (2S)−4′,5,6,7-tetrahydroxyflavanone 6-O-β-D-glucoside, 5,7,4′-trihydroxy-6-methoxyflavone3-O-β-D-rutinoside, kaempferol-3-O-β-D-glucoside, kaempferol-3-O-rutinoside, (2S)−4′,5,7,8-tetrahydroxy-flavanone-8-O-βD-glucoside, 6-hydroxykaempferol-3,6,7-tri-O-β-D-glucoside, and kaempferol-3-O-β-D-glucosyl-(1→2)-β-D-glucoside	It is distributed all over the world, such as in India, Mexico, America, Spain, Australia, and China with the following benefits: Promotes blood circulation and removes the stasis relieves pain treats headache and dizziness protects liver and relieves jaundice [42]	Medicinal liquor Decoction Pill, granule, capsule [42]

**Table 2 plants-11-01835-t002:** List of real-time PCR primer sequences.

Gene	Forward Primer	Reverse Primer
GAPDH	5′-CATGAGAAGTATGACAACAGCCT-3′	5′-AGTCCTTCCACGATACCAAAGT-3′
PCSK9	5′-GCTGAGCTGCTCCAGTTTCT-3′	5′-AATGGCGTAGACACCCTCAC-3′
LDLR	5′-AGTTGGCTGCGTTAATGTGA-3′	5′-TGATGGGTTCATCTGACCAGT-3′
HMGCR	5′-TGATTGACCTTTCCAGAGCAAG-3′	5′-CTAAAATTGCCATTCCACGAGC-3′

**Table 3 plants-11-01835-t003:** The grid position and grid size of the targeted protein.

Gene	Grid Position	Grid Size
PCSK9	34.025 × 23.492 × 25.638	110 × 82 × 126
HMGCR	73.702 × 0.468 × 18.849	122 × 78 × 126

**Table 4 plants-11-01835-t004:** Compounds identified in water-extracted *C. tinctorius*.

S. No.	RT	Name of the Compound	Molecular Formulae	MW	Peak Area (%)
1	6.10	D-Alanine, N-propargyloxycarbonyl-, isohexyl ester	C_13_H_21_NO_4_	255	3.14
2	7.72	4H-Pyran-4-one, 2,3-dihydro-3,5-dihydroxy-6-methyl-	C_6_H_8_O_4_	144	8.56
3	9.05	Acetic anhydride	C_4_H_6_O_3_	102	5.72
4	9.38	Benzofuran, 2,3-dihydro-	C_8_H_8_O	120	23.24
5	11.14	Cyclohexasiloxane, dodecamethyl-	C_12_H_36_O_6_Si_6_	444	6.96
6	14.59	Sucrose	C_12_H_22_O_11_	342	6.08
7	14.97	3,5-Dimethoxy-4-hydroxytoluene	C_9_H_12_O_3_	168	2.46
8	15.22	3-Isopropoxy-1,1,1,7,7,7-hexamethyl-3,5,5-tris(trimethylsiloxy)tetrasiloxane	C_18_H_52_O_7_Si_7_	576	13.73
9	16.45	2,4-Di-tert-butylphenol	C_14_H_22_O	206	4.68
10	16.83	Methyl 4-O-acetyl-2,3,6-tri-O-ethyl-.alpha.-d-galactopyranoside	C_15_H_28_O_7_	320	2.57
11	18.99	3,4-Dihydroxyphenylglycol, 4TMS derivative	C_20_H_42_O_4_Si_4_	458	8.94
12	22.27	3-Isopropoxy-1,1,1,7,7,7-hexamethyl-3,5,5-tris(trimethylsiloxy)tetrasiloxane-Dup1	C_18_H_52_O_7_Si_7_	576	4.79
13	25.21	3-Isopropoxy-1,1,1,7,7,7-hexamethyl-3,5,5-tris(trimethylsiloxy)tetrasiloxane-Dup2	C_18_H_52_O_7_Si_7_	576	2.71
14	27.89	Heptasiloxane, 1,1,3,3,5,5,7,7,9,9,11,11,13,13-tetradecamethyl-	C_18_H_44_O_6_Si_7_	504	1.79
15	29.52	Ethanol, 2,2′-(dodecylimino)bis-	C_16_H_35_NO_2_	273	2.34
16	39.48	Heptacosane	C_27_H_56_	380	1.36
17	41.25	Octacosane	C_28_H_58_	394	0.92

**Table 5 plants-11-01835-t005:** Compounds identified in the water-extracted *C. fenestratum*.

S. No.	RT	Name of the Compound	Molecular Formulae	MW	Peak Area (%)
1	6.72	Tert.-butylaminoacrylonitryl	C_7_H_12_N_2_	124	1.67
2	7.22	N-(Trimethylsilyl)pyridin-4-amine	C_8_H_14_N_2_Si	166	0.36
3	7.75	4H-Pyran-4-one, 2,3-dihydro-3,5-dihydroxy-6-methyl-	C_6_H_8_O_4_	144	0.22
4	8.79	Catechol	C_6_H_6_O_2_	110	0.9
5	9.06	Acetic anhydride	C_4_H_6_O_3_	102	0.49
6	10.67	Hydroquinone	C_6_H_6_O_2_	110	0.33
7	11.17	Cyclohexasiloxane, dodecamethyl-	C_12_H_36_O_6_Si_6_	444	0.38
8	12.67	Phenol, 2,6-dimethoxy-	C_8_H_10_O_3_	154	1.72
9	13.92	Benzaldehyde, 3-hydroxy-4-methoxy-	C_8_H_8_O_3_	152	0.24
10	15.22	3-Isopropoxy-1,1,1,7,7,7-hexamethyl-3,5,5-tris(trimethylsiloxy)tetrasiloxane	C_18_H_52_O_7_Si_7_	576	0.32
11	16.09	beta.-D-Glucopyranose, 1,6-anhydro-	C_6_H_10_O_5_	162	0.78
12	16.45	2,4-Di-tert-butylphenol	C_14_H_22_O	206	0.93
13	16.6	2-Methoxy-6-methoxycarbonyl-4-pyrone	C_8_H_8_O_5_	184	0.12
14	16.72	Benzoic acid, 4-hydroxy-3-methoxy-, methyl ester	C_9_H_810_O_4_	182	0.22
15	16.83	Methyl 4-O-acetyl-2,3,6-tri-O-ethyl-.alpha.-d-galactopyranoside	C_15_H_28_O_7_	320	2.1
16	16.96	2-Propanone, 1-(4-hydroxy-3-methoxyphenyl)-	C_10_H_12_O_3_	180	0.7
17	17.81	Megastigmatrienone	C_13_H_18_O	190	0.31
18	18.25	Megastigmatrienone-Dup1	C_13_H_18_O	190	0.99
19	19.02	Tetraacetyl-d-xylonic nitrile	C_14_H_17_NO_9_	343	27.92
20	19.31	Megastigmatrienone-Dup2	C_13_H_18_O	190	4.26
21	19.63	d-Gala-l-ido-octonic amide	C_8_H_17_NO_8_	255	0.19
22	19.81	2,6-Dimethoxyhydroquinone	C_8_H_10_O_4_	170	1.47
23	19.98	Benzaldehyde, 4-hydroxy-3,5-dimethoxy-	C_9_H_10_O_4_	182	3.35
24	20.28	d-Gala-l-ido-octonic amide-Dup1	C_8_H_17_NO_8_	255	9.75
25	20.64	Inositol, 1-deoxy-	C_6_H_12_O_5_	164	15.58
26	20.73	Inositol, 1-deoxy--Dup1	C_6_H_12_O_5_	164	9.31
27	21.11	3,4-Dihydrocoumarin, 4,4-dimethyl-6-hydroxy-	C_11_H_12_O_3_	192	0.14
28	21.75	(E)-4-(3-Hydroxyprop-1-en-1-yl)-2-methoxyphenol	C_10_H_12_O_3_	180	1.14
29	22.36	Benzoic acid, 4-hydroxy-3,5-dimethoxy-, methyl ester	C_10_H_12_O_5_	212	0.24
30	26.84	trans-Sinapyl alcohol	C_11_H_14_O_4_	210	1.66
31	29.52	Ethanol, 2,2′-(dodecylimino)bis-	C_16_H_35_NO_2_	273	0.66
32	35.65	Hentriacontane	C_31_H_64_	436	0.39
33	37.8	Octacosane, 2-methyl-	C_29_H_60_	408	0.54
34	39.48	Heptacosane	C_27_H_56_	380	0.69
35	40.48	Octacosane, 2-methyl-Dup1	C_29_H_60_	408	0.72
36	41.25	Hentriacontane-Dup1	C_31_H_64_	436	0.66
37	41.45	Doxepin	C_19_H_21_NO	279	0.11
38	42.02	Tetratetracontane	C_44_H_90_	618	0.36
39	42.31	1,4-Methano-2H-cyclopent[d]oxepin-2,5(4H)-dione, 6-[(dimethylamino)methyl]hexahydro-8a-hydroxy-5a-methyl-9-(1-methylethyl)-, [1R-(1.alpha.,4.alpha.,5a.alpha.,6.beta.,8a.alpha.,9S*)]-	C_17_H_27_NO_4_	309	0.66
40	42.66	Thieno[2,3-b]pyridine, 3-amino-2-(3,3-dimethyl-3,4-dihydroisoquinolin-1-yl)-4,6-dimethyl-	C_20_H_21_N_3_S	335	5.87
41	42.84	Octacosane	C_28_H_58_	394	0.28
42	46.12	1(4H)-naphthalenone, 4-[[4-(diethylamino)phenyl]imino]-2-hydroxy-	C_20_H_20_N_2_O_2_	320	0.21
43	49.16	Olean-12-en-28-oic acid, 3-hydroxy-, methyl ester, (3.beta.)-	C_31_H_50_O_3_	470	0.14

**Table 6 plants-11-01835-t006:** Compounds are identified in water-extracted *Z. officinale*.

S. No	RT	Name of the Compound	Molecular Formulae	MW	Peak Area (%)
1	5.45	3(2H)-Furanone, 4-hydroxy-5-methyl-	C_5_H_6_O_3_	114	0.55
2	6.13	Maltol	C_6_H_10_O_3_	126	2.78
3	6.73	Tert.-butylaminoacrylonitryl	C_7_H_12_N_2_	124	2.17
4	7.45	2-Propanamine, N-methyl-N-nitroso-	C_4_H_10_N_2_O	102	0.23
5	7.75	4H-Pyran-4-one, 2,3-dihydro-3,5-dihydroxy-6-methyl-	C_6_H_8_O_4_	144	2.14
6	8.76	Catechol	C_6_H_6_O_2_	110	0.8
7	9.13	Decanal	C_10_H_20_O	156	1.64
8	10.65	Cyclobuta[1,2:3,4]dicyclooctene, hexadecahydro-	C_16_H_28_	220	0.44
9	11.17	Cyclohexasiloxane, dodecamethyl-	C_12_H_36_O_6_Si_6_	444	0.89
10	11.79	2-Methoxy-4-vinylphenol	C_9_H_10_O_2_	150	0.48
11	14.09	10-Methyl-8-tetradecen-1-ol acetate	C_17_H_32_O_2_	268	0.53
12	14.72	2-Formyl-9-[.beta.-d-ribofuranosyl]hypoxanthine	C_11_H_12_N_4_O_6_	296	4.37
13	14.93	Cyclopentanecarboxaldehyde	C_6_H_10_O	98	0.52
14	15.22	3-Isopropoxy-1,1,1,7,7,7-hexamethyl-3,5,5-tris(trimethylsiloxy)tetrasiloxane	C_18_H_52_O_7_Si_7_	576	0.48
15	15.83	trans-Sesquisabinene hydrate	C_15_H_26_O	222	0.40
16	15.9	Benzene, 1-(1,5-dimethyl-4-hexenyl)-4-methyl-	C_15_H_22_	202	2.8
17	16.15	Octanal, 7-hydroxy-3,7-dimethyl-	C_10_H_20_O_2_	172	0.26
18	16.24	(1S,5S)-2-Methyl-5-((R)-6-methylhept-5-en-2-yl)bicyclo[3.1.0]hex-2-ene	C_15_H_24_	204	9.06
19	16.39	Alpha.-Farnesene	C_15_H_24_	204	2.1
20	16.46	Phenol, 2,5-bis(1,1-dimethylethyl)-	C_14_H_22_O	206	3.71
21	16.54	Beta.-Bisabolene	C_15_H_24_	204	2.33
22	16.81	3-Cyclohexene-1-methanol, 2-hydroxy-.alpha.,.alpha.,4-trimethyl-	C_10_H_8_O_2_	170	1.14
23	16.95	(1S,5S)-4-Methylene-1-((R)-6-methylhept-5-en-2-yl)bicyclo[3.1.0]hexane	C_15_H_24_	204	3.77
24	17.54	2-Furanmethanol, 5-ethenyltetrahydro-.alpha.,.alpha.,5-trimethyl-, cis-	C_10_H_18_O_2_	170	1.05
25	18.07	4-(1-Hydroxyallyl)-2-methoxyphenol	C_10_H_12_O_3_	180	1.39
26	18.58	Ethyl N-(o-anisyl)formimidate	C_10_H_13_NO_2_	179	0.49
27	18.99	Ethyl .alpha.-d-glucopyranoside	C_8_H_16_O_6_	208	3.1
28	19.7	2-Butanone, 4-(4-hydroxy-3-methoxyphenyl)-	C_11_H_14_O_3_	194	38.21
29	20.65	4-(3,4-Dimethoxyphenyl)butan-2-one	C_12_H_16_O_3_	208	0.17
30	20.86	(1R,2R,4S,6S,7S,8S)-8-Isopropyl-1-methyl-3-methylenetricyclo[4.4.0.02,7]decan-4-ol	C_15_H_24_O	220	0.26
31	23.26	cis-Z-.alpha.-Bisabolene epoxide	C_15_H_24_O	220	0.49
32	23.59	2-Naphthalenemethanol, decahydro-.alpha.,.alpha.,4a-trimethyl-8-methylene-, [2R-(2.alpha.,4a.alpha.,8a.beta.)]-	C_8_H_26_O	222	0.47
33	24.41	trans-Z-.alpha.-Bisabolene epoxide	C_15_H_24_O	220	0.43
34	25.48	Hexadecanoic acid, methyl ester	C_17_H_32_O_2_	270	0.2
35	29.52	Ethanol, 2,2′-(dodecylimino)bis-	C_16_H_35_NO_2_	273	0.89
36	31	(E)-1-(4-Hydroxy-3-methoxyphenyl)dec-3-en-5-one	C_17_H_24_O_3_	276	2.08
37	32.25	1-(4-Hydroxy-3-methoxyphenyl)dec-4-en-3-one	C_17_H_24_O_3_	276	5.89
38	35.44	(E)-4-(2-(2-(2,6-Dimethylhepta-1,5-dien-1-yl)-6-pentyl-1,3-dioxan-4-yl)ethyl)-2-methoxyphenol	C_27_H_42_O_4_	430	0.35
39	35.85	(3R,5S)-1-(4-Hydroxy-3-methoxyphenyl)decane-3,5-diyl diacetate	C_21_H_42_O_4_	380	0.69
40	39.74	1-(4-Hydroxy-3-methoxyphenyl)tetradec-4-en-3-one	C_21_H_32_O_3_	332	0.25

**Table 7 plants-11-01835-t007:** Compounds identified in ethanolic-extracted *C. tinctorius*.

S. No.	RT	Name of the Compound	Molecular Formulae	MW	Peak Area (%)
1	5.45	3(2H)-Furanone, 4-hydroxy-5-methyl-	C_5_H_6_O_3_	114	2.82
2	5.62	Acetic anhydride	C_4_H_6_O_3_	102	1.41
3	5.77	.gamma.-Dodecalactone	C_12_H_22_O_2_	198	4.31
4	6.13	Maltol	C_6_H_6_O_3_	126	4.2
5	6.74	Cyclopentanol	C_5_H_10_O	86	6.74
6	7.45	2-Propanamine, N-methyl-N-nitroso-	C_4_H_10_N_2_O	102	2.18
7	7.75	4H-Pyran-4-one, 2,3-dihydro-3,5-dihydroxy-6-methyl-	C_6_H_8_O_4_	144	7.76
8	7.87	4H-Pyran-4-one, 2,3-dihydro-3,5-dihydroxy-6-methyl--Dup1	C_6_H_8_O_4_	144	4.84
9	8.37	2H-Pyran, 3,4-dihydro-	C_5_H_8_O	84	2.42
10	8.58	5,8,11,14-Eicosatetraenoic acid, phenylmethyl ester, (all-Z)-	C_27_H_38_O_2_	394	0.69
11	8.76	Catechol	C_6_H_6_O_2_	110	4.27
12	9.06	Acetamide, N-[4-(4-nitrobenzylidenamino)-3-furazanyl]-	C_11_H_9_N_5_O_4_	275	3.43
13	9.4	Benzofuran, 2,3-dihydro-	C_8_H_8_O	120	3.51
14	9.61	5-Hydroxymethylfurfural	C_6_H_6_O_3_	126	1.33
15	10.67	Hydroquinone	C_6_H_6_O_2_	110	0.82
16	10.97	2-Butanone, 4-(ethylthio)-	C_6_H_12_OS	132	0.97
17	11.15	Cyclohexasiloxane, dodecamethyl-	C_12_H_36_O_6_Si_6_	444	3.31
18	11.77	2-Methyl-9-.beta.-d-ribofuranosylhypoxanthine	C_11_H_14_N_4_O_5_	282	2.04
19	12.66	Phenol, 2,6-dimethoxy-	C_8_H_10_O_3_	154	0.43
20	13.31	DL-Proline, 5-oxo-, methyl ester	C_6_H_9_NO_3_	143	1.76
21	13.9	4-Methyl(trimethylene)silyloxyoctane	C_12_H_26_OSi	214	1.72
22	14.16	3,7-Diacetamido-7H-s-triazolo[5,1-c]-s-triazole	C_7_H_9_N_7_O_2_	223	2.54
23	14.89	l-Pyrrolid-2-one, N-carboxyhydrazide	C_5_H_9_N_3_O_2_	143	6.13
24	14.97	Guanosine	C_10_H_13_N_5_O_5_	283	6.58
25	15.22	3-Isopropoxy-1,1,1,7,7,7-hexamethyl-3,5,5-tris(trimethylsiloxy)tetrasiloxane	C_18_H_52_O_7_ Si_7_	576	1.29
26	16.45	2,4-Di-tert-butylphenol	C_14_H_22_O	206	1.41
27	18.72	d-Glycero-d-ido-heptose	C_7_H_14_O_7_	210	1.42
28	19.44	3-Deoxy-d-mannonic acid	C_6_H_12_O_6_	180	7.85
29	19.68	d-Glycero-d-ido-heptose-Dup1	C_7_H_14_O_7_	210	3.94
30	19.87	2-Methyl-9-.beta.-d-ribofuranosylhypoxanthine-Dup1	C_11_H_14_N_4_O_5_	282	2.17
31	22.29	Heptasiloxane, 1,1,3,3,5,5,7,7,9,9,11,11,13,13-tetradecamethyl-	C_14_H_44_O_6_Si_7_	504	0.33
32	25.48	Hexadecanoic acid, methyl ester	C_17_H_34_O_2_	270	0.76
33	29.52	Ethanol, 2,2′-(dodecylimino)bis-	C_16_H_35_NO_2_	273	1.2
34	32.25	Heptacosane	C_27_H_56_	380	0.86
35	39.48	Heptacosane-Dup1	C_27_H_56_	380	1.45
36	40.34	9-Octadecenamide, (Z)-	C_18_H_35_NO	281	0.76
37	41.25	Heptacosane-Dup2	C_27_H_56_	380	0.35

**Table 8 plants-11-01835-t008:** Compounds identified in the ethanolic-extracted *C. fenestratum*.

S. No	RT	Name of the Compound	Molecular Formulae	MW	Peak Area (%)
1	6.11	3-Acetylthymine	C_7_H_8_N_2_O_3_	168	0.26
2	6.72	Tert.-butylaminoacrylonitryl	C_7_H_12_N_2_	124	1.1
3	7.22	4-Isopropylbenzenethiol, S-methyl-	C_10_H_14_S	166	0.34
4	7.75	4H-Pyran-4-one, 2,3-dihydro-3,5-dihydroxy-6-methyl-	C_6_H_8_O_4_	144	0.15
5	8.78	Catechol	C_6_H_8_O_2_	110	0.8
6	9.06	1-[3-(4-Bromophenyl)-2-thioureido]-1-deoxy-b-d-glucopyranose 2,3,4,6-tetraacetate	C_21_H_25_BrN_2_O_9_S	560	0.26
7	11.15	Cyclohexasiloxane, dodecamethyl-	C_12_H_36_O_6_Si_6_	444	0.09
8	11.79	2-Methoxy-4-vinylphenol	C_9_H_10_O_2_	150	0.17
9	12.68	Phenol, 2,6-dimethoxy-	C_9_H_10_O_3_	154	0.09
10	13.31	2-Pyrrolidinone, 5-(cyclohexylmethyl)-	C_11_H_19_NO	181	0.18
11	13.92	Benzaldehyde, 3-hydroxy-4-methoxy-	C_8_H_8_O_3_	152	0.15
12	15.23	3-Isopropoxy-1,1,1,7,7,7-hexamethyl-3,5,5-tris(trimethylsiloxy)tetrasiloxane	C_18_H_52_O_7_Si_7_	576	0.4
13	16.11	.beta.-D-Glucopyranose, 1,6-anhydro-	C_6_H_10_O_5_	162	0.67
14	16.46	2,4-Di-tert-butylphenol	C_14_H_22_O	206	0.37
15	16.59	2-Methoxy-6-methoxycarbonyl-4-pyrone	C_8_H_8_O_5_	184	0.1
16	16.73	Benzoic acid, 4-hydroxy-3-methoxy-, methyl ester	C_9_H_10_O_4_	182	0.11
17	16.84	Methyl 4-O-acetyl-2,3,6-tri-O-ethyl-.alpha.-d-galactopyranoside	C_15_H_28_O_7_	320	0.19
18	16.97	2-Propanone, 1-(4-hydroxy-3-methoxyphenyl)-	C_10_H_12_O_3_	180	0.26
19	17.8	Megastigmatrienone	C_13_H_18_O	190	0.19
20	18.24	Megastigmatrienone-Dup1	C_13_H_18_O	190	0.86
21	18.71	3,4,5-Trimethoxyphenol	C_9_H_12_O_4_	184	1.91
22	19.01	Cyclopropanetetradecanoic acid, 2-octyl-, methyl ester	C_26_H_50_O_2_	394	7.19
23	19.18	Tetraacetyl-d-xylonic nitrile	C_14_H_17_NO_9_	343	9.47
24	19.32	Megastigmatrienone-Dup2	C_13_H_18_O	190	11.58
25	19.81	2-Oxa-3-azabicyclo[4.4.0]dec-3-ene, 5-methyl-1-trimethylsilyloxy-, N-oxide	C_12_H_23_NO_3_Si	257	2
26	19.98	Benzaldehyde, 4-hydroxy-3,5-dimethoxy-	C_9_H_10_O_4_	182	3
27	20.07	.alpha.-l-Mannose semicarbazone pentaacetate	C_18_H_25_N_3_O_12_	475	1.48
28	20.28	d-Gala-l-ido-octonic amide	C_8_H_17_NO_8_	255	7
29	20.6	Shikimic acid	C_7_H_10_O_5_	174	4.84
30	20.9	(E)-2,6-Dimethoxy-4-(prop-1-en-1-yl)phenol	C_11_H_14_O_3_	194	8.69
31	21.13	Inositol, 1-deoxy-	C_6_H_12_O_5_	164	6.02
32	21.46	Inositol, 1-deoxy--Dup1	C_6_H_12_O_5_	164	15.44
33	21.75	(E)-4-(3-Hydroxyprop-1-en-1-yl)-2-methoxyphenol	C_10_H_12_O_3_	180	1.26
34	22.28	3-Isopropoxy-1,1,1,7,7,7-hexamethyl-3,5,5-tris(trimethylsiloxy)tetrasiloxane-Dup1	C_18_H_52_O_7_Si_7_	576	0.2
35	22.37	Benzoic acid, 4-hydroxy-3,5-dimethoxy-, methyl ester	C_10_H_12_O_5_	212	0.72
36	22.82	4-Hydroxy-4a,8-dimethyl-3-methylene-3,3a,4,4a,7a,8,9,9a-octahydroazuleno[6,5-b]furan-2,5-dione	C_15_H_18_O_4_	262	0.12
37	25.48	Hexadecanoic acid, methyl ester	C_17_H_34_O_2_	270	0.19
38	26.83	trans-Sinapyl alcohol	C_11_H_14_O_4_	210	0.43
39	27.89	1,3-Dioxolo[4,5-g]isoquinolin-5(6H)-one, 7,8-dihydro-	C_10_H_9_NO_3_	191	0.1
40	28.68	9,12-Octadecadienoic acid, methyl ester, (E,E)-	C_19_H_34_O_2_	294	0.08
41	28.8	9-Octadecenoic acid (Z)-, methyl ester	C_19_H_36_O_2_	296	0.15
42	29.52	Ethanol, 2,2′-(dodecylimino)bis-	C_16_H_35_NO_2_	273	0.25
43	40.47	7-Isoquinolinol, 1,2,3,4-tetrahydro-1-[(3-hydroxy-4-methoxyphenyl)methyl]-6-methoxy-2-methyl-, (S)-	C_19_H_23_NO_4_	329	0.09
44	41.05	Corydine	C_20_H_23_NO_4_	341	0.06
45	41.45	Ethylamine, 2-((p-bromo-.alpha.-methyl-.alpha.-phenylbenzyl)oxy)-N,N-dimethyl-	C_18_H_22_BrNO	347	0.06
46	42.3	1-Undecanamine, N,N-dimethyl-	C_13_H_29_N	199	0.32
47	42.67	Thieno[2,3-b]pyridine, 3-amino-2-(3,3-dimethyl-3,4-dihydroisoquinolin-1-yl)-4,6-dimethyl-	C_20_H_21_N_3_S	335	4.26
48	44.57	Berbine, 13,13a-didehydro-9,10-dimethoxy-2,3-(methylenedioxy)-	C_20_H_19_NO_4_	337	1.3
49	44.69	Ergosta-5,22-dien-3-ol, acetate, (3.beta.,22E)-	C_30_H_48_O_2_	440	0.28
50	45.31	Thalictricavine	C_21_H_23_NO_4_	353	0.12
51	45.42	.beta.-Sitosterol	C_29_H_50_O	414	0.21
52	46.15	1(4H)-naphthalenone, 4-[[4-(diethylamino)phenyl]imino]-2-hydroxy-	C_20_H_20_N_2_O_2_	320	1.29
53	49.19	Olean-12-en-28-oic acid, 3-hydroxy-, methyl ester, (3.beta.)-	C_31_H_50_O_3_	470	2.15
54	50.18	Urs-12-en-28-oic acid, 3-hydroxy-, methyl ester, (3.beta.)-	C_31_H_50_O_3_	470	0.23
55	50.35	Urs-12-en-28-oic acid, 3-hydroxy-, methyl ester, (3.beta.)-Dup1	C_31_H_50_O_3_	470	0.78

**Table 9 plants-11-01835-t009:** Compounds are identified in ethanolic-extracted *Z. officinale*.

S. No.	RT	Name of the Compound	Molecular Formulae	MW	Peak Area (%)
1	9.12	Decanal	C_10_H_20_O	156	3.1
2	10.18	2,6-Octadien-1-ol, 3,7-dimethyl-, (Z)-	C_10_H_18_O	154	0.91
3	15.91	Benzene, 1-(1,5-dimethyl-4-hexenyl)-4-methyl-	C_15_H_22_	202	1.28
4	16.23	1,3-Cyclohexadiene, 5-(1,5-dimethyl-4-hexenyl)-2-methyl-, [S-(R*,S*)]-	C_15_H_24_	204	3.79
5	16.4	.alpha.-Farnesene	C_15_H_24_	204	1.19
6	16.55	.beta.-Bisabolene	C_15_H_24_	204	0.93
7	16.94	Cyclohexene, 3-(1,5-dimethyl-4-hexenyl)-6-methylene-, [S-(R*,S*)]-	C_15_H_24_	204	2.03
8	17.74	Nerolidol	C_15_H_26_O	222	0.89
9	18.07	4-(1-Hydroxyallyl)-2-methoxyphenol	C_10_H_12_O_3_	180	1.35
10	19.8	Butan-2-one, 4-(3-hydroxy-2-methoxyphenyl)-	C_11_H_14_O_3_	194	33.27
11	20.11	2-Naphthalenemethanol, decahydro-.alpha.,.alpha.,4a-trimethyl-8-methylene-, [2R-(2.alpha.,4a.alpha.,8a.beta.)]-	C_15_H_26_O	222	1.4
12	20.66	(1S,2R,5R)-2-Methyl-5-((R)-6-methylhept-5-en-2-yl)bicyclo[3.1.0]hexan-2-ol	C_15_H_26_O	222	0.99
13	20.87	1H-3a,7-Methanoazulen-5-ol, octahydro-3,8,8-trimethyl-6-methylene-	C_15_H_24_O	220	1.63
14	23.27	cis-Z-.alpha.-Bisabolene epoxide	C_15_H_24_O	220	1.83
15	24.43	trans-Z-.alpha.-Bisabolene epoxide	C_15_H_24_O	220	0.83
16	24.55	Acetic acid, 3-hydroxy-6-isopropenyl-4,8a-dimethyl-1,2,3,5,6,7,8,8a-octahydronaphthalen-2-yl ester	C_17_H_26_O_3_	278	0.65
17	25.48	Hexadecanoic acid, methyl ester	C_17_H_34_O_2_	270	0.42
18	31.02	(E)-1-(4-Hydroxy-3-methoxyphenyl)dec-3-en-5-one	C_17_H_24_O_3_	276	4.96
19	31.2	3-Decanone, 1-(4-hydroxy-3-methoxyphenyl)-	C_17_H_26_O_3_	278	1.5
20	32.33	1-(4-Hydroxy-3-methoxyphenyl)dec-4-en-3-one	C_17_H_24_O_3_	276	24.37
21	35.48	(E)-4-(2(2-(2,6-Dimethylhepta-1,5-dien-1-yl)-6-pentyl-1,3-dioxan-4-yl)ethyl)-2-methoxyphenol	C_27_H_42_O_4_	430	1.13
22	35.87	1-(4-Hydroxy-3-methoxyphenyl)dodec-4-en-3-one	C_19_H_28_O_3_	304	5.23
23	38.64	(E)-1-(4-Hydroxy-3-methoxyphenyl)tetradec-3-en-5-one	C_21_H_32_O_3_	332	0.74
24	39.74	1-(4-Hydroxy-3-methoxyphenyl)tetradec-4-en-3-one	C_21_H_32_O_3_	332	3.24
25	40.14	1-(4-Hydroxy-3-methoxyphenyl)tetradecane-3,5-dione	C_21_H_32_O_4_	348	0.35
26	42.79	(E)-4-(2(2-(2,6-Dimethylhepta-1,5-dien-1-yl)-6-pentyl-1,3-dioxan-4-yl)ethyl)-2-methoxyphenol-Dup1	C_27_H_42_O_4_	430	0.61
27	45.43	.beta.-Sitosterol	C_29_H_50_O	414	1.35

**Table 10 plants-11-01835-t010:** The pocket binding site of PCSK9.

No.	Binding Site	Amino Acid
1	Strong No. 1	ILE:154, PRO:155, ASN:157, LEU:158, GLU:159, ARG:160, ILE:161, THR:162, PRO:163, ARG:165, TYR:166, ARG:167, ARG:237, ASP:238, ALA:239, GLY:240, VAL:241, ALA:242, LYS:243, GLY:244, GLY:394, ILE:395, ALA:397, MET:398, MET:399, LEU:400, SER:401, ALA:402, GLU:403, LEU:406, ARG:414, PHE:418, ALA:443, LEU:444, PRO:445, PRO:446, SER:447, THR:448, HIS:449, GLY:450, ALA:451
2	Strong No. 2	ALA:68:A, LYS:69:A, GLY:292, TYR:293, SER:294, ARG:295, LEU:297, ASN:298, ALA:299, ALA:300, CYS:301, GLN:302, ARG:303, LEU:304, ALA:305, ARG:306, ALA:307, GLY:308, VAL:309, THR:313, ASP:321, ALA:322, CYS:323, LEU:324, TYR:325, SER:326, PRO:327, ALA:328, SER:329, ALA:330, PRO:331, GLU:332, VAL:333, ILE:334, THR:335, GLY:356, ARG:357, CYS:358, VAL:359, ASP:360, LEU:361, THR:407, LEU:408, ALA:409, GLU:410, ARG:412, GLN:413, ILE:416, HIS:417, SER:419, ALA:420, LYS:421, ASP:422, VAL:423, ILE:424, ASN:425, GLU:426, ALA:427, PHE:429, GLU:431, ASP:432, GLN:433, ARG:434, VAL:435, LEU:436, THR:437, PRO:438, ASN:439, LEU:440, CYS:457, ARG:458, THR:459, VAL:460, TRP:461, SER:462, ALA:463, HIS:464, SER:465, GLY:466, ALA:471, THR:472, ALA:473, ILE:474, ALA:475, ARG:476, CYS:477, ALA:478, PRO:479, ASP:480, GLU:481, GLU:482, LEU:483, PHE:489, ARG:491, GLU:501, GLY:505, LYS:506, LEU:507, VAL:508, ARG:510, VAL:520, TYR:521, ALA:522, ILE:523, ARG:525, CYS:526, GLU:620, GLN:621, THR:623, VAL:624, ALA:625, CYS:626, TYR:648, ALA:649, VAL:650, ASP:651, ASN:652, THR:653, CYS:654, VAL:655, ARG:657
3	Strong No. 3	CYS:486, SER:487, SER:488, GLY:493, LYS:494, ARG:495, ARG:496, GLY:497, GLU:498, ALA:514, PHE:515, ARG:549, LEU:559, GLY:561, CYS:562, SER:563, SER:564, HIS:565, TRP:566, GLU:567, VAL:568, GLU:569, ASP:570, GLN:584, PRO:585, ASN:586, GLN:587, CYS:588, VAL:589, GLY:590, HIS:591, ARG:592, GLU:593, ALA:594, SER:595, ILE:596, HIS:597, LYS:609, VAL:610, LYS:611, GLU:612, GLY:634, CYS:635, SER:636, ALA:637, LEU:638, PRO:639, SER:642, HIS:643, VAL:644, LEU:645, GLY:646, ALA:647, TYR:648, VAL:656, ALA:671, ALA:674, VAL:675, ALA:676, ILE:677
4	Medium	GLU:159, ARG:160, ILE:161, THR:162, PRO:163, PRO:164, ARG:165, TYR:166, ASP:343, GLU:403, GLN:413, ARG:414, ILE:416, HIS:417, PHE:418, SER:419, ALA:420, LYS:421, ASP:422, VAL:423, LEU:440, VAL:441, ALA:442, ALA:443, LEU:444, PRO:445, PRO:446, SER:447, THR:448, HIS:449, GLY:450, ALA:451, GLY:452, TRP:453, GLN:454, LEU:455, PHE:456, CYS:457, ARG:458, ARG:525, LEU:606, LYS:611, ALA:625, CYS:626, GLU:627, GLU:628, GLY:629, TRP:630, THR:631, LEU:632, VAL:650, ASP:651, ASN:652, THR:653, CYS:679, ARG:680, SER:681, ARG:682

**Table 11 plants-11-01835-t011:** Energy binding and phytochemical inhibition constants of herbal extracts with water at the binding sites of PCSK9 from ArgusLab and Autodock analysis and quantification of each compound through GC-MS/MS analysis.

No.	Herb	Compound Name	GC-MS/MS	ArgusLab	Autodock	
			% Peak Area	Binding Energy (kcal/mol)	Binding Energy (kcal/mol)	Inhibition Constant (Ki)
1		Alirocumab (Positive control)		−7.59	−5.61	77.42 µM
2	*C. tinctorius*	Benzofuran, 2,3-dihydro-	23.24	−8.90	−5.43	104.25 µM
3		3-Isopropoxy-1,1,1,7,7,7-hexamethyl-3,5,5-tris(trimethylsiloxy)tetrasiloxane	21.23	N/B	−5.47	97.4 µM
4		3,4-Dihydroxyphenylglycol, 4TMS derivative	8.94	−8.63	−7.54	2.96 µM
5		4H-Pyran-4-one, 2,3-dihydro-3,5-dihydroxy-6-methyl-	8.56	−6.19	−6.99	7.46 µM
6		Cyclohexasiloxane, dodecamethyl-	6.96	−8.34	−7.88	1.69 µM
7	*C. fenestratum*	d-Gala-l-ido-octonic amide	9.94	−7.15	−6.46	18.3 µM
8		Inositol, 1-deoxy-	24.89	−8.33	−7.30	4.48 µM
9		Tetraacetyl-d-xylonic nitrile	27.92	−8.26	−6.76	11.05 µM
10		Thieno[2,3-b]pyridine, 3-amino-2-(3,3-dimethyl-3,4-dihydroisoquinolin-1-yl)-4,6-dimethyl-	5.87	−11.14	−10.15	36.5 nM
11		Megastigmatrienone	5.56	−10.83	−7.87	1.7 µM
12	*Z. officinale*	2-Butanone, 4-(4-hydroxy-3-methoxyphenyl)-	38.21	−8.73	−7.66	2.42 µM
13		(1S,5S)-2-Methyl-5-((R)-6-methylhept-5-en-2-yl)bicyclo[3.1.0]hex-2-ene	9.06	−10.26	−7.25	4.82 µM
14		1-(4-Hydroxy-3-methoxyphenyl)dec-4-en-3-one	5.89	−10.32	−8.35	754.12 nM
15		2-Formyl-9-[.beta.-d-ribofuranosyl]hypoxanthine	4.37	−7.62	−10.79	12.4 nM
16		(1S,5S)-4-Methylene-1-((R)-6-methylhept-5-en-2-yl)bicyclo[3.1.0]hexane	3.77	−11.26	−7.40	3.78 µM

N/B: No suitable ligand poses were discovered.

**Table 12 plants-11-01835-t012:** Energy binding and phytochemical inhibition constants of herbal extracts with ethanol at the binding sites of PCSK9 from ArgusLab and Autodock analysis and quantification of each compound through GC-MS/MS analysis.

No.	Herb	Compound Name	GC-MS/MS	ArgusLab	Autodock	
			% Peak Area	Binding Energy (kcal/mol)	Binding Energy (kcal/mol)	Inhibition Constant (Ki)
1		Alirocumab (Positive control)		−7.59	−5.61	77.42 µM
2	*C. tinctorius*	Cyclopentanol	6.74	−8.29	−5.27	137.36 µM
3		3-Deoxy-d-mannonic acid	7.85	−7.43	−6.93	8.27 µM
4		Guanosine	6.58	−7.47	−11.31	5.16 nM
5		l-Pyrrolid-2-one, N-carboxyhydrazide	6.13	−7.27	−7.38	3.89 µM
6		4H-Pyran-4-one, 2,3-dihydro-3,5-dihydroxy-6-methyl-	12.60	−6.19	−6.99	7.46 µM
7	*C. fenestratum*	(E)-2,6-Dimethoxy-4-(prop-1-en-1-yl)phenol	8.69	−8.76	−7.51	3.11 µM
8		Cyclopropanetetradecanoic acid, 2-octyl-, methyl ester	7.19	−12.56	−5.14	169.81 µM
9		Megastigmatrienone	12.63	−10.83	−7.87	1.7 µM
10		Inositol, 1-deoxy-	21.46	−8.33	−7.30	4.48 µM
11		Thieno[2,3-b]pyridine, 3-amino-2-(3,3-dimethyl-3,4-dihydroisoquinolin-1-yl)-4,6-dimethyl-	4.26	−11.14	−10.15	36.5 nM
12	*Z. officinale*	1,3-Cyclohexadiene, 5-)1,5-dimethyl-4-hexenyl)-2-methyl-, [S-(R*,S*)]	3.79	−10.91	−7.36	4.0 µM
13		1-(4-Hydroxy-3-methoxyphenyl)dodec-4-en-3-one	5.23	−11.29	−8.69	428.1 nM
14		(E)-1-(4-Hydroxy-3-methoxyphenyl)dec-3-en-5-one	4.96	−10.40	−8.8	351.96 nM
15		Butan-2-one, 4-(3-hydroxy-2-methoxyphenyl)-	33.27	−8.25	−7.44	3.54 µM
16		1-(4-Hydroxy-3-methoxyphenyl)dec-4-en-3-one	24.37	−10.32	−8.35	754.12 nM

**Table 13 plants-11-01835-t013:** Energy binding and phytochemical inhibition constants of herbal extracts with water at the binding sites of HMGR from ArgusLab and Autodock analysis and quantification of each compound through GC-MS/MS analysis.

No.	Herb	Compound Name	GC-MS/MS	ArgusLab	Autodock	
			% Peak Area	Binding Energy (kcal/mol)	Binding Energy (kcal/mol)	Inhibition Constant (Ki)
1		Lovastatin (Positive control)		−9.23012	−8.55	540.36 nM
2	*C. tinctorius*	Benzofuran, 2,3-dihydro-	23.24	−8.19673	−5.91	46.78 μM
3		3-Isopropoxy-1,1,1,7,7,7-hexamethyl-3,5,5-tris(trimethylsiloxy)tetrasiloxane	21.23	N/B	−5.15	168.03 μM
4		3,4-Dihydroxyphenylglycol, 4TMS derivative	8.94	−7.66333	−6.60	14.6 μM
5		4H-Pyran-4-one, 2,3-dihydro-3,5-dihydroxy-6-methyl-	8.56	−6.64198	−7.22	5.07 μM
6		Cyclohexasiloxane, dodecamethyl-	6.96	−7.98578	−7.59	2.75 μM
7	*C. fenestratum*	d-Gala-l-ido-octonic amide	9.94	−7.64931	−5.85	51.27 μM
8		Inositol, 1-deoxy-	24.89	−8.28603	−7.34	4.15 μM
9		Tetraacetyl-d-xylonic nitrile	27.92	−7.88168	−6.49	17.48 μM
10		Thieno[2,3-b]pyridine, 3-amino-2-(3,3-dimethyl-3,4-dihydroisoquinolin-1-yl)-4,6-dimethyl-	5.87	−10.0154	−7.75	2.07 μM
11		Megastigmatrienone	5.56	−9.73578	−6.04	37.12 μM
12	*Z. officinale*	2-Butanone, 4-(4-hydroxy-3-methoxyphenyl)-	38.21	−9.35038	−5.90	47.54 μM
13		(1S,5S(-2-Methyl-5-((R)-6-methylhept-5-en-2-yl)bicyclo[3.1.0]hex-2-ene	9.06	−10.7714	−5.82	54.27 μM
14		1-(4-Hydroxy-3-methoxyphenyl)dec-4-en-3-one	5.89	−10.5172	−6.10	33.53 μM
15		2-Formyl-9-[.beta.-d-ribofuranosyl]hypoxanthine	4.37	−7.52531	−7.96	1.47 μM
16		(1S,5S)-4-Methylene-1-((R)-6-methylhept-5-en-2-yl)bicyclo[3.1.0]hexane	3.77	−10.1426	−5.41	108.68 μM

**Table 14 plants-11-01835-t014:** Energy binding and phytochemical inhibition constants of herbal extracts with ethanol at the binding sites of HMGR from ArgusLab and Autodock analysis and quantification of each compound through GC-MS/MS analysis.

No.	Herb	Compound Name	GC-MS/MS	ArgusLab	Autodock	
			% Peak Area	Binding Energy (kcal/mol)	Binding Energy (kcal/mol)	Inhibition Constant (Ki)
1		Lovastatin (Positive control)		−9.23012	−8.55	540.36 nM
2	*C. tinctorius*	Cyclopentanol	6.74	−8.37591	−4.72	345.87 μM
3		3-Deoxy-d-mannonic acid	7.85	−7.71546	−4.19	845.72 μM
4		Guanosine	6.58	−8.31259	−7.77	2 μM
5		l-Pyrrolid-2-one, N-carboxyhydrazide	6.13	−7.38878	−6.99	7.54 μM
6		4H-Pyran-4-one, 2,3-dihydro-3,5-dihydroxy-6-methyl-	12.60	−6.64198	−7.22	5.07 μM
7	*C. fenestratum*	(E)-2,6-Dimethoxy-4-(prop-1-en-1-yl)phenol	8.69	−8.90424	−6.69	12.52 μM
8		Cyclopropanetetradecanoic acid, 2-octyl-, methyl ester	7.19	−11.2679	−3.62	2.22 mM
9		Megastigmatrienone	12.63	−9.73578	−6.04	37.12 μM
10		Inositol, 1-deoxy-	21.46	−8.28603	−7.34	4.15 μM
11		Thieno[2,3-b]pyridine, 3-amino-2-(3,3-dimethyl-3,4-dihydroisoquinolin-1-yl)-4,6-dimethyl-	4.26	−10.0154	−7.75	2.07 μM
12	*Z. officinale*	1,3-Cyclohexadiene, 5-(1,5-dimethyl-4-hexenyl)-2-methyl-, [S-(R*,S*)]-	3.79	−10.5606	−5.80	56.41 μM
13		1-(4-Hydroxy-3-methoxyphenyl)dodec-4-en-3-one	5.23	−10.681	−5.43	104.75 μM
14		(E)-1-(4-Hydroxy-3-methoxyphenyl)dec-3-en-5-one	4.96	−10.2192	6.04	37.24 μM
15		Butan-2-one, 4-(3-hydroxy-2-methoxyphenyl)-	33.27	−8.67751	−5.68	69.13 μM
16		1-(4-Hydroxy-3-methoxyphenyl)dec-4-en-3-one	24.37	−10.5172	−6.10	33.53 μM

**Table 15 plants-11-01835-t015:** Energy binding and phytochemical inhibition constants of herbal extracts with water at the binding sites of SREBP2 from ArgusLab and Autodock analysis and quantification of each compound through GC-MS/MS analysis.

No.	Herb	Compound Name	GC MS/MS	ArgusLab	Autodock	
			% Peak Area	Binding Energy (kcal/mol)	Binding Energy (kcal/mol)	Inhibition Constant (Ki)
1		Metformin (Positive control)		−5.87716	−5.56	84.25 μM
2	*C. tinctorius*	Benzofuran, 2,3-dihydro-	23.24	−8.62431	−5.08	189.21 μM
3		3-Isopropoxy-1,1,1,7,7,7-hexamethyl-3,5,5-tris(trimethylsiloxy)tetrasiloxane	21.23	N/B	−5.85	51.15 μM
4		3,4-Dihydroxyphenylglycol, 4TMS derivative	8.94	−7.48581	−4.38	615.35 μM
5		4H-Pyran-4-one, 2,3-dihydro-3,5-dihydroxy-6-methyl-	8.56	−6.69362	−6.49	17.62 μM
6		Cyclohexasiloxane, dodecamethyl-	6.96	−7.32463	−7.03	7.06 μM
7	*C. fenestratum*	d-Gala-l-ido-octonic amide	9.94	−7.16144	−5.95	43.16 μM
8		Inositol, 1-deoxy-	24.89	−7.83301	−6.89	8.83 μM
9		Tetraacetyl-d-xylonic nitrile	27.92	−7.59524	−5.13	173.8 μM
10		Thieno[2,3-b]pyridine, 3-amino-2-(3,3-dimethyl-3,4-dihydroisoquinolin-1-yl)-4,6-dimethyl-	5.87	−9.68843	−9.91	54.25 nM
11		Megastigmatrienone	5.56	−11.7348	−7.37	3.97 μM
12	*Z. officinale*	2-Butanone, 4-(4-hydroxy-3-methoxyphenyl)-	38.21	−8.93613	−7.39	3.82 μM
13		(1S,5S)-2-Methyl-5-((R)-6-methylhept-5-en-2-yl)bicyclo[3.1.0]hex-2-ene	9.06	−12.9835	−7.32	4.3 μM
14		1-(4-Hydroxy-3-methoxyphenyl)dec-4-en-3-one	5.89	−11.3944	−8.62	476.42 nM
15		2-Formyl-9-[.beta.-d-ribofuranosyl]hypoxanthine	4.37	−7.4906	−8.69	425.74 nM
16		(1S,5S)-4-Methylene-1-((R)-6-methylhept-5-en-2-yl)bicyclo[3.1.0]hexane	3.77	−12.7577	−7.41	3.72 μM

**Table 16 plants-11-01835-t016:** Energy binding and phytochemical inhibition constants of herbal extracts with ethanol at the binding sites of SREBP2 from ArgusLab and Autodock analysis and quantification of each compound through GC-MS/MS analysis.

No.	Herb	Compound Name	GC MS/MS	ArgusLab	Autodock	
			% Peak Area	Binding Energy (kcal/mol)	Binding Energy (kcal/mol)	Inhibition Constant (Ki)
1		Metformin (Positive control)		−5.87716	−5.56	84.25 μM
2	*C. tinctorius*	Cyclopentanol	6.74	−7.35609	−4.51	498.19 μM
3		3-Deoxy-d-mannonic acid	7.85	−7.11679	−5.37	115.85 μM
4		Guanosine	6.58	−7.51631	−9.56	99.06 nM
5		l-Pyrrolid-2-one, N-carboxyhydrazide	6.13	−6.78964	−6.51	16.87 μM
6		4H-Pyran-4-one, 2,3-dihydro-3,5-dihydroxy-6-methyl-	12.60	−6.69362	−6.49	17.62 μM
7	*C. fenestratum*	(E)-2,6-Dimethoxy-4-(prop-1-en-1-yl)phenol	8.69	−9.32055	−7.45	3.49 μM
8		Cyclopropanetetradecanoic acid, 2-octyl-, methyl ester	7.19	−11.2105	−4.97	227.42 μM
9		Megastigmatrienone	12.63	−11.7348	−7.37	3.97 μM
10		Inositol, 1-deoxy-	21.46	−7.83301	−6.89	8.83 μM
11		Thieno[2,3-b]pyridine, 3-amino-2-(3,3-dimethyl-3,4-dihydroisoquinolin-1-yl)-4,6-dimethyl-	4.26	−9.68843	−9.91	54.25 nM
12	*Z. officinale*	1,3-Cyclohexadiene, 5-(1,5-dimethyl-4-hexenyl)-2-methyl-, [S-(R*,S*)]-	3.79	−11.7619	−7.05	6.75 μM
13		1-(4-Hydroxy-3-methoxyphenyl)dodec-4-en-3-one	5.23	−11.602	−4.88	265.67 μM
14		(E)-1-(4-Hydroxy-3-methoxyphenyl)dec-3-en-5-one	4.96	−10.7057	−6.15	30.88 μM
15		Butan-2-one, 4-(3-hydroxy-2-methoxyphenyl)-	33.27	−9.25557	−5.93	45.14 μM
16		1-(4-Hydroxy-3-methoxyphenyl)dec-4-en-3-one	24.37	−11.3944	−4.88	265.67 μM

## Data Availability

The datasets used and/or analyzed during the current study are available from the corresponding author Komgrit Eawsakul (komgrit.ea@wu.ac.th) on reasonable request.

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
