# Peer review of "The Cholesterol-Modulating Effect of the New Herbal Medicinal Recipe from Yellow Vine (Coscinium fenestratum (Goetgh.)), Ginger (Zingiber officinale Roscoe.), and Safflower (Carthamus tinctorius L.) on Suppressing PCSK9 Expression to Upregulate LDLR Expression in HepG2 Cells"

_plants, 2022, doi:10.3390/plants11141835_

Round 1

Reviewer 1 Report

The manuscript is of great interest since the topic of manuscript is devoted to the extremely actual problem of current medicine such as cardiovascular disease prevention. The article is in scope of journal and includes the results of the study aimed to develop a combination of medicinal plants possessing anti-hyperlipidemic action through inhibiting PCSK9 expression and to evaluate its cholesterol-lowering effect and cytotoxicity. The manuscript is well-organized, written in a good English, and may be recommended for publication in Plants after minor correction.

Line 597: Despite of the fact, that PCSK9 inhibition is beneficial for lipid reduction, “statin 597 therapy alone will not be able to lower blood lipids” needs a revision.

Minor spelling correction is required.

Reviewer 2 Report

An article by a large group of authors from different countries entitled: "The cholesterol-modulating effect of the new herbal medicinal recipe from Yellow vine (Coscinium fenestratum (Goetgh.)), Ginger (Zingiber officinale Roscoe.), and Safflower (Carthamus tinctorius L.) on suppressing PCSK9 expression to upregulate LDLR expression in HepG2 Cells" is devoted to topical issues regarding the use of plants for the control of cholesterol in human blood. High levels of cholesterol are a significant risk factor for atherosclerosis and cardiovascular disease. Reducing blood lipid profile may aid in the treatment of high levels of cholesterol-related diseases and disorders, including metabolic syndrome.

The main provision of this work is the rationale for the control of cholesterol in the blood in connection with diseases of the human cardiovascular system. Blood cholesterol levels of total cholesterol and low-density lipoprotein (LDL) cholesterol are both major risk factors for coronary heart disease (CHD). Reduced total and LDL cholesterol levels have been shown to decrease the risk of coronary heart disease.

The methodological part of the work is presented in sufficient detail. Modern approaches were used to identify the biological effect of plant extracts (or preparations).

In this study, the lowering cholesterol activity of three plants, including safflower (Carthamus tinctorius), yellow vine (Coscinium fenestratum), and ginger (Zingiber officinale), as well as the potential molecular mechanisms involved in their lowering cholesterol activity, were investigated in the human liver cell line HepG2 by using molecular docking and RT-qPCR. Furthermore, proved that combining these plants by making three parts yellow vine (primary herb), two parts ginger (support herb), and one-part safflower (coloring herb) significantly reduced lipid accumulation in hepatocytes by investigating oil red O staining.

The revealed effects of the use of aqueous and alcoholic (ethanol) extracts on lipid reduction are convincingly shown and proven. Therefore, the authors need to prove with great justification that it was the objects of research they chose that were taken not by chance, but purposefully (which is why it is important to show the use of these plants for human food).

Comments and/or recommendations to the authors. When describing the species, one should give information about each in a more detailed form. Namely: the degree of knowledge of the chemical composition of primary and secondary metabolites accumulated by plants; give in tabular form information and use of these plants in different countries (with links to publications) in traditional medicine, as well as drawing attention to the features and specifics of the use of herbal preparations in different countries.

The table should indicate one of the options for the name of the plants: either folk, or (correctly) - only the Latin name of the plants (the journal is scientific - therefore it is important to use the scientific names of plants).

At the beginning of the article, all names of plants (folk, possibly different for different countries) and the correct scientific (Latin) name in full (with the author of the species) should be given. Further in the text and in the tables, only one Latin name should be given, but without indicating the author of the species.

Reviewer 3 Report

In this manuscript, the authors developed a recipe (safflower, yellow vine, and ginger) that combined molecular docking, GC-MS/MS, and real-time PCR to identify potential 35 PCSK9 inhibitors for herb ratio determination, and evaluate the cholesterol-lowering through a 37 PCSK9 inhibitory mechanism of three herbs. The review comments are summarized as follows:

1. In the introduction section, the information about ethno-pharmacological significance, usage and so on of the plant should be presented.

2. As for sample preparation, the author should present the reason for selection of extraction method, since diverse approaches obtain different chemical compounds.

3. Standard deviation for the data of determination of maximum dose for HepG2 was lost in the result part,please check and revise.

4. Figures 5, 6 and 7 are in poor quality. The authors need to revise the figures 5, 6 and 7 at higher quality.

5. The structure-activity relationship between active compounds by GC-MS/MS analysis and corresponding activities should be further discussed in the discussion part.

Round 2

Reviewer 3 Report

The authors have addressed most of the comments raised, and the revised version is improved.  Thus, acceptance of the work is suggested.

Author Response

We thank the reviewer for the positive comment on our previous revision and no changes are required.